# Graph Neural Networks Designed for Different Graph Types: A Survey

**Josephine M. Thomas**[*]                                                                                      *jthomas@uni-kassel.de*
*GAIN – Graphs in Artificial Intelligence and Machine Learning*
*Intelligent Embedded Systems*
*University of Kassel, Germany*

**Alice Moallemy-Oureh**[*]                                                                                      *amoallemy@uni-kassel.de*
*GAIN – Graphs in Artificial Intelligence and Machine Learning*
*Intelligent Embedded Systems*
*University of Kassel, Germany*

**Silvia Beddar-Wiesing**[*]                                                                                      *s.beddarwiesing@uni-kassel.de*
*GAIN – Graphs in Artificial Intelligence and Machine Learning*
*Intelligent Embedded Systems*
*University of Kassel, Germany*

**Clara Holzhüter**[*]                                                                                      *clara.juliane.holzhueter@iee.fraunhofer.de*
*GAIN – Graphs in Artificial Intelligence and Machine Learning*
*Fraunhofer Institute for Energy Economics and Energy System Technology (IEE)*
*Kassel, Germany*

**Reviewed on OpenReview:** *https://openreview.net/forum?id=h4BYtZ79uy*

## Abstract

Graphs are ubiquitous in nature and can therefore serve as models for many practical but also theoretical problems. For this purpose, they can be defined as many different types which suitably reflect the individual contexts of the represented problem. To address cutting-edge problems based on graph data, the research field of Graph Neural Networks (GNNs) has emerged. Despite the field's youth and the speed at which new models are developed, many recent surveys have been published to keep track of them. Nevertheless, it has not yet been gathered which GNN can process what kind of graph types. In this survey, we give a detailed overview of already existing GNNs and, unlike previous surveys, categorize them according to their ability to handle different graph types and properties. We consider GNNs operating on static and dynamic graphs of different structural constitutions, with or without node or edge attributes. Moreover, we distinguish between GNN models for discrete-time or continuous-time dynamic graphs and group the models according to their architecture. We find that there are still graph types that are not or only rarely covered by existing GNN models. We point out where models are missing and give potential reasons for their absence.

## 1 Introduction

Over the last decades, neural networks (NNs) have become increasingly important. Their development dates back to the early 1940s (Anderson & Rosenfeld, 1988) [1]. With increasing computational power and the possibility of utilizing Deep Learning (DL), their applications have reached most parts of society, from detecting cancer (McKinney et al., 2020) to playing computer games (Ibarz et al., 2018; Silver et al.,

---

[*]All authors contributed equally.
[1]Anderson & Rosenfeld (1988) provides a historical overview up to the end of the 1980s.

2018). Nevertheless, classical NNs are limited to Euclidean data. Given the rising amount of non-Euclidean data (Bronstein et al., 2017) and the fact that graphs are a suitable mathematical representation for many theoretical and practical problems, several authors started investigating NNs on particular graph problems (Cimikowski & Shope, 1996; Lai et al., 1994) or so-called "structures" (Sperduti, 1997; Sperduti & Starita, 1997) in the 90s. With an ever-increasing amount of graph data available (see, e.g., repositories Rossi & Ahmed (2015), or OGB Hu et al. (2020)) in many applications (e.g., traffic (Ma et al., 2020; Rossi & Ahmed, 2015), citation (Feng et al., 2019; Ioannidis et al., 2019; Ren et al., 2020; Tran & Tran, 2020), biological or medical (La Gatta et al., 2020; Wang et al., 2020; Yadati et al., 2019; Zitnik et al., 2018), social (Pareja et al., 2020; Rossi et al., 2020; Trivedi et al., 2019), recommendation (Sankar et al., 2020; Wang et al., 2021; Yang et al., 2020)), so-called Graph Neural Networks (GNNs) have become a thriving research field.

Therefore, many surveys have recently conducted intensive research on GNN models, e.g., Barros et al. (2021); Kazemi et al. (2020); Skarding et al. (2021); Zhou et al. (2020). However, most GNN models are either limited to a specific graph type or developed to address particular problems. E.g., Hier-GNN Chen et al. (2022) is developed especially for hierarchical graphs, MXMNet Zhang et al. (2020a) for multiplex graphs, and EpiGNN La Gatta et al. (2020) focuses on learning the evolution of an epidemic. On the other hand, real-world graphs are diverse. In many cases, they contain heterogeneous nodes or edges and evolve dynamically. One example of a heterogeneous graph is a power grid representation in which the nodes could have different types, such as "solar power plants", "wind parks", or "nuclear power plants". An example of a dynamic graph is a social network with time-changing nodes and the connections among them. However, no comprehensive overview is available that investigates which graph types are addressed by existing GNN models. Since the graph type plays a vital role in choosing a model to solve a graph problem, it is essential to provide an overview of the latest collection of GNNs.

This survey aims to fill this gap by providing an outline of GNNs for all graph types and pointing out the absent GNN models for static and dynamic graphs. As a comprehensive overview of the different graph types is missing, the first contribution of this survey consists of the definition and overview of these. It covers basic structural graph types (e.g., directed, multi-, heterogeneous, or hypergraphs) for static and dynamic graphs in discrete and continuous-time and the so-called semantic graph types (e.g., cyclic, regular, and bipartite graphs). This categorization approach is advantageous because some GNN models are restricted to specific graph properties. The second contribution is an analysis of which graph types can be handled by currently available GNN models. As a third contribution, we group the investigated GNN models by their architecture in the main part. The final contribution consists in analyzing what graph types cannot be handled by current GNN models, including explanations for these gaps.

Due to the vast amount of publications in the field, this survey cannot cover all existing models. Therefore, this survey aims to cover the most important models and list only one or two models for each graph type or property to illustrate the existence of at least one model. The following criteria determine the importance of the models for the choice: 1) Up-to-dateness of the model, 2) relevance of the model concerning the number of citations and its use as a baseline in other publications, 3) the generality of the model (e.g., that it is not only applicable to a particular domain), 4) explicitness in addressing the listed graph properties, and 5) simplicity of the model (e.g., if two models fulfill the same task, priority is given to the simpler one). The individual reason for the choice of each model can be found in the appendix in Tab.8.

This paper is structured as follows. Sec. 2 contains related work. In Sec. 3, the considered graphs and their properties are defined in 3.1, while preliminary definitions concerning GNNs are given in 3.2. Sections 4 to 7 constitute the central part of the paper and deal with GNN models focusing on structural graph properties (Sec. 4), dynamic graph properties (Sec. 5), semantic graph properties (Sec. 6) and combined or other GNN models (Sec. 7). Here, each section contains a table showing which graph types and properties are addressed by existing GNN models, a description of the applied GNN techniques, and an evaluation of why current models might not cover specific properties. Note that many models have the same acronym in the respective publication. Therefore we altered some of them to distinguish the models and improve readability. Finally, Sec. 8 concludes the work and points out future challenges. The mathematical notation used throughout this work can be found in Sec. 9, Tab. 7.

## 2 Related Work

Several surveys that review GNNs concerning different aspects have been proposed over the last few years. Multiple surveys provide a more detailed overview of specific types of methods, such as convolutional GNNs (Gama et al., 2020; Zhang et al., 2018; 2019b), GNNs using attention mechanisms (Lee et al., 2019), or Baysian GNNs (Shi et al., 2021). Furthermore, many existing surveys focus on specific application areas (Jiang & Luo, 2022; Shlomi et al., 2020; Wu et al., 2022), such as natural language processing (Wu et al., 2021), combinatorial optimization (Peng et al., 2021), or power systems (Liao et al., 2021). Other publications reviewing GNN models concentrate on specific aspects such as explainability (Yuan et al., 2021), or the expressive power of GNNs (Sato, 2020). Unlike these publications, we provide a more general survey, which is neither limited to particular types of methods or aspects nor explicit application fields.

Cai et al. (2018) provide a broad survey of graph embedding techniques, including methods apart from deep learning, such as matrix factorization or graph kernels, similar to (Cui et al., 2018; Goyal & Ferrara, 2018; Hamilton et al., 2017). In Bronstein et al. (2017), an overview of deep learning methods applicable to non-Euclidian data is provided. The survey does not only focus on graphs but aims to cover methods of geometric deep learning in general, including its applications, challenges, and future directions. Concerning GNNs, it primarily surveys convolutional methods. However, the aforementioned surveys do not cover methods for dynamic graphs. In contrast, (Wu et al., 2020) covers spatial-temporal GNNs, convolutional methods, recurrent GNNs, and graph autoencoders. The investigated methods are grouped according to these categories. Similarly, (Zhang et al., 2020b) review models by the type of GNN they apply. However, these categories differ from thosse in Wu et al. (2020) such that instead of spatial-temporal GNNs, graph reinforcement learning and adversarial methods are discussed. Both methods only partially cover dynamic graph models.

Further publications such as Barros et al. (2021); Kazemi et al. (2020); Skarding et al. (2021) explicitly focus on models for dynamic graphs. Skarding et al. (2021) further group the reviewed models concerning the encoded type of dynamics (e.g., node dynamic, edge-growing) and the applied methods. While Barros et al. (2021) and Skarding et al. (2021) survey models for dynamic graphs only, Kazemi et al. (2020) also reviews several static methods. However, the corresponding chapter of this survey aims to understand better the basic concepts for static graphs, which can be extended to dynamic graphs, rather than reviewing methods for static graphs. None of the abovementioned surveys categorizes the reviewed methods for different graph types and their semantic properties. The only survey that explicitly investigates GNN models concerning the graph types is Zhou et al. (2020). However, it does not consider all graph types covered in this survey since we provide a more fine-grained distinction of different graph properties. Moreover, in Zhou et al. (2020) the authors focus on the pipeline of designing a GNN, including identifying the graph type and additional network modules such as pooling or sampling. Accordingly, it takes a different point of view and reviews GNN models amongst other modules, which can be integrated into a deep learning pipeline.

Our contribution is a detailed overview of existing GNNs and their categorization into certain types of methods, but more importantly, the types of graphs they can process. Unlike many existing surveys, we consider static and dynamic graphs. Moreover, we group the corresponding dynamic GNNs into discrete-time and continuous-time dynamic models while considering the node and edge attributes and the graph's structure.

## 3 Foundations

The application of graphs takes place in many different fields. This is because of the high degree of freedom in designing a graph and, thus, in representing information. Therefore, many different graph types have been developed and extended over time. This section defines graph types and properties in detail to give the reader a comprehensive insight into all graph types and associated GNNs in detail and presents them in order. Readers familiar with the different graph types and properties may omit this section and go on to the following section, Sec. 4, for an overview of existing GNN models and architectures.

For the remainder of this section, the reader is assumed to have basic knowledge of analysis and linear algebra (see, for example, Abbott (2001); Strang (1993)). A table containing the most frequently used notation can be found in Sec. 9.

### 3.1 Graphs And Their Properties

At first, the considered graph properties and graph types have to be defined to survey for which graph types and properties GNN models exist. These definitions are given here, as well as some graph-related terms which are needed throughout the paper.

We distinguish between structural and semantic graph properties. The graph structure is defined only by the mathematical objects that make up the graph, i.e., node, edge, and attribute sets. The so-called structural properties can be deduced from these sets, e.g., whether the graph is directed or attributed. Semantic properties, in contrast, do not affect the mathematical representation. They result from interpreting the graph, e.g., whether it is cyclic or a tree. Some GNNs specialize in such properties since they frequently occur in real-world applications. All definitions concerning structural properties are taken from Thomas et al. (2021) and given here for the reader's comfort.

In the following, elementary graph types are defined. They form the basis for all graphs to which neural networks have already been applied or might be applied in the future.

**Definition 3.1 (Static Graphs: Elementary)**
  1. A **directed (simple) graph** is a tuple $G = (\mathcal{V}, \mathcal{E})$ containing a set of nodes $\mathcal{V} \subset \mathbb{N}$ and a set of directed edges given as tuples $\mathcal{E} \subseteq \mathcal{V} \times \mathcal{V}$.
  2. A **(generalized) directed hypergraph** is a tuple $G = (\mathcal{V}, \mathcal{E})$ with nodes $\mathcal{V} \subset \mathbb{N}$ and hyperedges

$$\mathcal{E} \subseteq \{(\mathsf{x}, f_i)_i \mid \mathsf{x} \subseteq \mathcal{V}, \, f_i : \mathsf{x} \to \mathbb{N}_0\}$$

that include a numbering map $f_i$ for the $i$-th edge $(\mathsf{x}, f)_i$ which indicates the order of the nodes in the (generalized) directed hyperedge. W.l.o.g. it can be assumed that the numbering is gap-free, so if there exists a node $u \in \mathsf{x}$ with $f(u) = k > 1$ then there will also exist a node $v$ s.t. $f(v) = k - 1$.

These graphs are called elementary because every other graph is a composition of them. In this sense, a **directed hypergraph** is a directed simple graph that simultaneously is a hypergraph. Since one can not only combine elementary graphs but also extend them with additional properties, in what follows, different types of graph properties are introduced. Namely the static structural, dynamic structural, and semantic properties.

**Definition 3.2 (Static Structural Graph Properties)**
An elementary graph $G = (\mathcal{V}, \mathcal{E})$ is called

  1. **undirected** if the edge directions are irrelevant, i.e.,
     - for directed graphs: if $(u, v) \in \mathcal{E}$ whenever $(v, u) \in \mathcal{E}$ for $u, v \in \mathcal{V}$. Then, the edges can be denoted as a set of sets instead of a set of tuples, namely

$$\mathcal{E} \subseteq \{\{u, v\} \mid u, v \in \mathcal{V}, u \neq v\} \cup \{\{u\} \mid u \in \mathcal{V}\}^2,$$

     - for directed hypergraphs: if $f_i : \mathsf{x} \to \{0\}$ for all $(\mathsf{x}, f_i)_i \in \mathcal{E}^3$. Abbreviated by $\mathcal{E} \subseteq \{\mathsf{x} \mid \mathsf{x} \subseteq \mathcal{V}\}$.
  2. **multigraph** if it is a multi-edge graph, i.e., the edges $\mathcal{E}$ are defined as a multiset[4], a multi-node graph, i.e., the node set $\mathcal{V}$ is a multiset, or both.
  3. **heterogeneous** if the nodes or edges can have different types (node or edge-heterogeneous). Mathematically, the type is appended to the nodes and edges. I.e., the node set is determined by $\mathcal{V} \subseteq \mathbb{N} \times \mathcal{S}$ with a node type set $\mathcal{S}$ and thus, a node $(v, s) \in \mathcal{V}$ is given by the node $v$ itself and its type $s$. The edges can be extended by a set $\mathcal{R}$ that describes their types, to $(e, r) \; \forall \, e \in \mathcal{E}$ of edge type $r \in \mathcal{R}$.
  4. **attributed** if the nodes $\mathcal{V}$ or edges $\mathcal{E}$ are equipped with node or edge attributes. These attributes are formally given by a node attribute function and an edge attribute function, respectively, i.e. $\alpha : \mathcal{V} \to \mathcal{A}$ and $\omega : \mathcal{E} \to \mathcal{W}$, where $\mathcal{A}$ and $\mathcal{W}$ are arbitrary attribute sets. In case there are only node attributes the graph is called **node-attributed** (or node labeled/node features), in case of just edge attributes it is called **edge-attributed** and if we have $\mathcal{W} \subseteq \mathbb{R}$ it is called **weighted**.

---

[2] the second set contains the set of self-loops
[3] $f_i(\mathsf{x}) = 0$ encodes that $\mathsf{x}$ is an undirected hyperedge
[4] A multiset is a set that can have entries which occur multiple times.

Fig. 1 shows examples for each graph type up to this point.

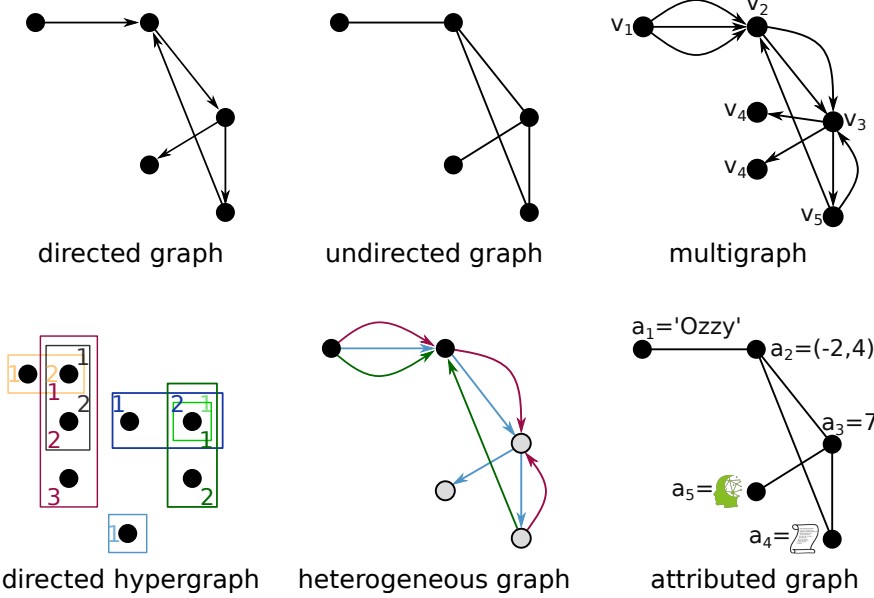

Figure 1: Visualization of different elementary static graph types.

The term **static** in these structural properties stands for the absence of temporal dependence. This means, in particular, that once the graph is given, it never changes with time. In contrast, the so-called (temporal) dynamic structural graph properties are listed in the following.

**Definition 3.3 (Dynamic Structural Graph Properties)**
A graph is called

1. **dynamic** if the graph structure or the graph properties are time dependent. In the following, the notation $G_i = (\mathcal{V}_i, \mathcal{E}_i)$, $t_i \in T$ is used, where $T$ is a set of (not necessarily equidistant) timestamps to emphasize the time-dependence and therefore the dynamics.
2. **growing** if it is dynamic and the node or edge sets evolve w.r.t. addition of new nodes and edges respectively. I.e., for all $t_i \in T$ it holds

$$\mathcal{V}_i \subseteq \mathcal{V}_{i+1} \quad \text{or} \quad \mathcal{E}_i \subseteq \mathcal{E}_{i+1}.$$

3. **shrinking** if it is dynamic and we just allow node or edge set evolution w.r.t. deletions of nodes and edges respectively. I.e., for all $t_i \in T$, it is

$$\mathcal{V}_i \supseteq \mathcal{V}_{i+1} \quad \text{or} \quad \mathcal{E}_i \supseteq \mathcal{E}_{i+1}.$$

4. **strictly growing/shrinking** if we consider only real inclusions in definition 2 and 3 above.
5. **structure-dynamic** if it is growing, shrinking or both simultaneously, i.e., in particular, the nodes $\mathcal{V}$ or edges $\mathcal{E}$ evolve over time due to additions or deletions of nodes or edges[5].
6. **attribute-dynamic** if the node or edge attribute function is time-dependent. Thus, we extend our notions of the attribute functions to $\alpha_i : \mathcal{V}_i \to \mathcal{A}$ and $\omega_i : \mathcal{E}_i \to \mathcal{W}$, for all $t_i \in T$.
7. **type-dynamic** if the graph type evolves over time. E.g., an undirected graph becomes directed from one to another time step.

Structurally, these dynamics describe different temporal behaviors of graphs. When processing dynamic graphs, they are typically defined either as discrete-time or continuous-time representations.

---

[5](Kazemi et al., 2020) also mentions splits and merges of nodes and edges. Obviously, these events are sequences of additions and deletions.

**Definition 3.4 (Dynamic Graph Representation)**

1. A dynamic graph in **discrete-time representation** is given by a set $\mathcal{G} = \{g_1, \ldots, g_k\}$ of graph snapshots $g_i$ at time steps $i = 1, \ldots, k$. Here, $g_i := (\mathcal{V}_i, \mathcal{E}_i)$ are static graphs with nodes $\mathcal{V}$ and edges $\mathcal{E}_i \subseteq \{(u, v) \mid u, v \in \mathcal{V}_i\}$.

2. A dynamic graph in **continuous-time representation** is defined by a set $G = \{g_{t_0}, \mathbb{E}\}$ containing an initial static graph at time stamp $t_0 \in \mathcal{T}$ and a set $\mathbb{E} = \{e_t, t \in \mathcal{T}\}$ of events encoding a structural or attribute change at time stamp $t > t_0 \in \mathcal{T}$.

Not all combined graphs are equally important in the literature and especially for GNNs. The following introduces some combined graph types of specific interest with proper names.

**Definition 3.5 (Combined Static Graphs)**

1. **Knowledge graphs** are defined in several ways. In Wang et al. (2021), they are defined as heterogeneous directed graphs, while in Yu et al. (2020) knowledge graphs are the same as edge-heterogeneous graphs. But there are also definitions that do not see a knowledge graph as a graph combined from the aforementioned types, see for example Ehrlinger & Wöß (2016) for an overview.

2. A **multi-relational graph** (Hamilton, 2020) is an edge-heterogeneous but node-homogeneous graph.

3. A **content-associated heterogeneous graph** is a heterogeneous graph with node attributes that correspond to heterogeneous data like, e.g., attributes, text or images (Zhang et al., 2019a).

4. A **multiplex graph/multi-channel graph** corresponds to an edge-heterogeneous graph with self-loops (Hamilton, 2020). Here, we have $k$ layers, where each layer consists of the same node set $\mathcal{V}$, but different edge sets $\mathcal{E}^{(k)}$. Additionally, inter-layer edges $\tilde{\mathcal{E}}$ exist between the same nodes across different layers.

5. A **spatio-temporal graph** is a multiplex graph where edges per each layer are interpreted as spatial edges and the inter-layer edges indicate temporal steps between a layer at time step $t$ and $t + 1$. They are called temporal edges (Kapoor et al., 2020).

*Remark.* All the combined properties mentioned in Def. 3.5 can occur in dynamic graphs as well.

Besides the structural properties, a graph can have semantic properties that do not explicitly change its structure but result from applying or interpreting the graph information. Some GNNs are limited or specialized to these properties defined in the following.

**Definition 3.6 (Semantic Graph Properties)**

An elementary graph $G = (\mathcal{V}, \mathcal{E})$ is called

1. **complete** if all pairwise different nodes are connected through an edge, i.e., $\mathcal{E} = \{(u, v) \in \mathcal{V} \times \mathcal{V} \mid u \neq v\}$.

2. $r$-**regular** if each node $v \in \mathcal{V}$ has $r \in \mathbb{N}$ neighbors, i.e.,

$$|\mathcal{N}(v)| := |\{u \in \mathcal{V} \mid (u, v) \in \mathcal{E}\}| = r.$$

3. **bipartite** if there exists a disjoint node decomposition into two sets $\mathcal{V} = \mathcal{U} \,\dot\cup\, \mathcal{W}$, such that the edges are of the form $\mathcal{E} \subseteq \mathcal{U} \times \mathcal{W}$.

4. **connected** if the graph is undirected and for all node pairs $v, w \in \mathcal{V}$ there is a path from $v$ to $w$ in $G$. An elementary graph is called **weakly connected** if the underlying undirected graph is connected and it is **strongly connected** if for all node pairs $v, w \in \mathcal{V}$ there is a directed path from $v$ to $w$ in $G$. Otherwise it is unconnected.

5. **cyclic** if it contains a cycle of length $k \in \mathbb{N}$, i.e., there exists a subgraph $H = (\{v_1, \ldots, v_k\}, \{e_1, \ldots, e_k\}) \subseteq G$, $v_i \in \mathcal{V}$, $e_i \in \mathcal{E} \;\forall i$, such that the series of nodes and edges $v_1, e_1, v_2, \ldots, v_k, e_k, v_1$ is a closed (directed) path called **(directed) cycle** of length $k$ with $v_i \neq v_j \;\forall i, j$. Otherwise, it is called **acyclic** or a **forest**.

6. **tree** if it is a connected forest. In case each node in the tree has at most two neighbors, it is called **binary tree** (Gessel & Stanley, 1995). A **polytree** is a directed graph whose underlying undirected graph is a tree (Dasgupta, 1999).

7. **level-$(l+1)$ hierarchical** w.r.t. a level-$l$ base graph $H = (\tilde{\mathcal{V}}, \tilde{\mathcal{E}})$ if one can find a complete partitioning of $H$ into $k \geq 1$ non-empty, connected sets of nodes $\tilde{\mathcal{V}}_1, .., \tilde{\mathcal{V}}_k$. Such that each set of nodes $\tilde{\mathcal{V}}_i \subseteq \tilde{\mathcal{V}}$ induces a subgraph $sub_i(H) = (\tilde{\mathcal{V}}_i, \tilde{\mathcal{E}}_i \subseteq \tilde{\mathcal{E}})$ with $\tilde{\mathcal{E}}_i = \{(v_1, v_2) \in \tilde{\mathcal{E}} \mid v_1, v_2 \in \tilde{\mathcal{V}}_i\}$. Each of these subgraphs, in turn, corresponds to a node in the hierarchical graph $G$. Edges in $G$ correspond to edges in $H$ between nodes $v_i, v_j$ of two different subgraphs $sub_i(H), sub_j(H)$ (Stoffel et al., 2008).

8. **scale-free**, if its node degree distribution $P(d)$ follows a power law $P(d)\ d^{-\gamma}$, where $\gamma$ typically lies within the range $2 < \gamma < 3$.

A (generalized) (directed) hypergraph $G = (\mathcal{V}, \mathcal{E})$ is called

9. **recursive**, if an edge can not only exist between nodes, but also between edges and in a recursive way. E.g. two edges $e_1$ and $e_2$ make up edge $e_{12}$, edges $e_3$ and $e_4$ make up edge $e_{34}$ and edge $e_5$ consists of edges $e_{12}$ and $e_{34}$. See definition 4 of an ubergraph in Joslyn & Nowak (2017) and figure 1a in Yadati (2020) for a visualization.

Fig. 2 shows examples for each semantic graph property by applying it to undirected graphs.

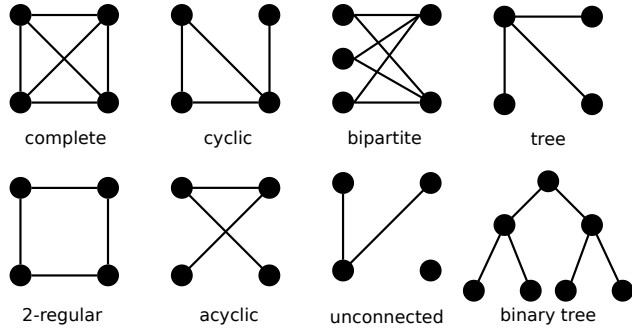

Figure 2: Semantic graph properties illustrated for undirected graphs.

In the following chapter, we introduce the basic architectures for GNNs that make up all the GNNs in this survey. In order to be able to describe these appropriately, we list some frequently occurring graph-related terms beforehand.

**Definition 3.7 (Graph related terms)**
Let $G \coloneqq (\mathcal{V}, \mathcal{E})$ be a graph.

1. The **degree of a node** $v \in \mathcal{V}$ is given by $\delta(v) = |\{e \in \mathcal{E} \mid v \in e\}|$. For directed graphs, the **out- or in-degree** of $v$ is the number of edges starting in $v$ or ending in $v$, respectively. The **degree of an edge** $e \in \mathcal{E}$ is determined by $|e|$, i.e., by the number of nodes in the edge.
2. The **graph Laplacian** or **Laplacian matrix** $\boldsymbol{L}$ is defined by $\boldsymbol{L} = \boldsymbol{D} - \boldsymbol{A}$, where $\boldsymbol{D}$ is the degree matrix and $\boldsymbol{A}$ the adjacency matrix. In Graph Convolutional Neural Networks, it is mostly used in a normalized version, e.g., the symmetric and normalized graph Laplacian $\boldsymbol{L}_{norm} = \tilde{\boldsymbol{D}}^{-\frac{1}{2}} \tilde{\boldsymbol{A}} \tilde{\boldsymbol{D}}^{-\frac{1}{2}}$, where $\tilde{\boldsymbol{A}}$ is the adjacency matrix with self connections and $\tilde{\boldsymbol{D}}$ the degree matrix with self-loops.
3. An entry $y_{i,j}$ of the **incidence matrix** $\boldsymbol{Y} \in \{0, 1\}^{|\mathcal{V}| \times |\mathcal{E}|}$ of a graph $G = (\mathcal{V}, \mathcal{E})$ is 1, if the node $i$ is incident to edge $j$, and 0 otherwise. For non-hypergraphs, the incidence matrix has exactly 2 entries per row that are non-zero. Let $\tilde{\mathcal{V}} \subseteq \mathcal{V}$ be a set of nodes. Then, the **induced subgraph of** $\tilde{\mathcal{V}}$ is defined by a graph $G(\tilde{\mathcal{V}}) = (\tilde{\mathcal{V}}, \tilde{\mathcal{E}})$ with edges $\tilde{\mathcal{E}} = \{e \in \mathcal{E} \mid e \in \tilde{\mathcal{V}} \times \tilde{\mathcal{V}}\}$ between the nodes of $\tilde{\mathcal{V}}$.
4. A **path** from $u \in \mathcal{V}$ to $v \in \mathcal{V}$, denoted by $p(u,v) \coloneqq e_1, \ldots, e_k \in \mathcal{E}$ is a sequence of edges, for which there is a sequence of nodes $(z_1, ..., z_{k+1})$ such that $e_i = (z_i, z_{i+1})$ for $i = 2, ..., k$ and $e_1 = (u, x)$ and $e_{k+1} = (y, v)$ for some $x, y \in \mathcal{V}$.
5. The **path length** of a path $p(u,v) = e_1, \ldots e_k$ denotes the sequence length, i.e., $\text{len}(p(u,v)) = k$.
6. A **random walk of length** $k$ is a path of length $k$ whose edges are selected iteratively and randomly.

### 3.2 GNN Preliminaries

GNNs define the adaptation of traditional NNs to graph data and aim to learn high-level representations of graphs in an end-to-end fashion by applying several network layers. They can be applied to all classical machine learning problems, such as classification, regression, or clustering, for entire graphs and subgraphs at a node or edge level. Each layer computes a new representation of the graph or its components. A typical procedure is to update the representation for the nodes in each layer by propagating information through

the graph. A task-specific prediction can then be made using the learned representation and a suitable decoder function. For node classification, e.g., a typical choice for the decoder is a standard MLP with a softmax activation as the output function. It maps the learned representation to a vector indicating the class probabilities for all nodes. At the edge level, a frequently considered task is link prediction which aims to predict the probability of the existence of an edge. The corresponding decoder is often implemented as a logistic regression classifier since the existence of an edge can be expressed as a two-class problem.

Different types of GNNs specify the computation of the node representation in the GNN layers. According to Bronstein et al. (2021), the following relation of GNN approaches applies:

$$\text{message-passing} \supseteq \text{attention} \supseteq \text{convolution}$$

Therefore, these are introduced one after the other, from the most general case to special ones. A visualization of the three GNN layer types is shown in figure 3. The Recurrent Neural Networks coexist with the message-passing and will be introduced afterward. Combined with GNNs, it is particularly relevant for dynamic graph learning problems due to its ability to model temporal data.

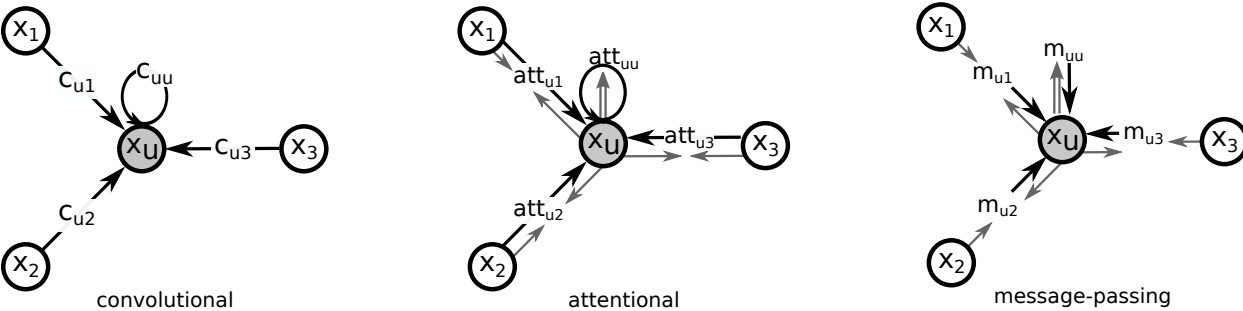

Figure 3: Visualization of the information propagation process in the different types of GNN layers for node $u$ and its neighbors. The idea for the figure is taken from Bronstein et al. (2021, Fig. 17). *Left:* In convolutional layers, the node features $x_v$ of the neighbors $v \in \{1, \ldots, 3\}$ of node $u$ are multiplied with a constant $c_{u,v}$ to form the message. *Middle:* In attention layers, this multiplier is computed via an attention mechanism $att_{uv} = att(x_u, x_v)$ between the source and the target nodes $u, v$. *Right:* In message-passing layers, the messages $m_{uv}$ are computed explicitly from the source and target node representations, i.e., $m_{uv} = \psi(x_u, x_v)$.

### 3.2.1 Message-Passing

Message-passing determines which and how much information is forwarded between two nodes, e.g., via their connecting edges. The resulting representation $\boldsymbol{h}_u$ of node $u$ is computed from the node representations $\boldsymbol{x}_v \ \forall v \in \mathcal{V}$:

$$\boldsymbol{h}_u = \phi\left(\boldsymbol{x}_u, \bigoplus_{v \in \mathcal{N}(u)} \psi(\boldsymbol{x}_u, \boldsymbol{x}_v)\right), \tag{1}$$

where $\psi$ is a learnable message function that assigns an information vector to the pair $u, v$. Typically, $\psi$ is defined as multiplication with a learnable weight matrix, and its output is denoted as a message. The aggregation $\oplus$ depicts the message-passing process on the graph, which in most cases is implemented as a non-parametric operation such as sum, mean, or maximum. $\mathcal{N}(u)$ denotes a neighborhood of node $u$ and $\phi$ is an activation function (Bronstein et al., 2021).

### 3.2.2 (Multi-head) Graph Attention

Graph attention is a special case of message-passing (Bronstein et al., 2021). Here, the message is computed by applying a learnable function $\psi$ to each neighboring node weighted by a so-called *attention* factor. Typically,

the function $\psi$ is shared across all neighbors, whereas the attention is computed individually for each node pair. The attention mechanism specifies the message-passing rule in the aggregation function as follows:

$$\boldsymbol{h}_u = \phi\left(\boldsymbol{x}_u, \bigoplus_{v \in \mathcal{N}(u)} \mathrm{att}(\boldsymbol{x}_u, \boldsymbol{x}_v)\psi(\boldsymbol{x}_v)\right), \tag{2}$$

where the attention function att is learnable and determines the effect of the message from neighbor $v$ with representation $\psi(\boldsymbol{x}_v)$ to the hidden representation $\boldsymbol{h}_u$ of node $u$. Additionally, the attention coefficients are normalized across all neighbors of the target node. Note that $att_k$, therefore, depends not only on $x_u$ and $x_v$ but also on all other neighbors of node $x_u$. Furthermore, if $\oplus$ is a sum, the aggregation is a linear combination considering feature-specific weights for the neighbors.

Multi-head attention extends the attention mechanism to $K$ different attention functions (Velickovic et al., 2018) and is determined by

$$\boldsymbol{h}_u = \mathop{\|}_{k \in [K]} \phi\left(\boldsymbol{x}_u, \bigoplus_{v \in \mathcal{N}(u)} \mathrm{att}_k(\boldsymbol{x}_u, \boldsymbol{x}_v)\psi(\boldsymbol{x}_v)\right), \tag{3}$$

where $\|$ denotes the concatenation operation. The $K$ different attention functions also called *attention heads* are computed independently. In Velickovic et al. (2018), an implementation of an attention mechanism is proposed. The according self-attention function $\mathrm{att} : \mathbb{R}^{\dim(\boldsymbol{h})} \times \mathbb{R}^{\dim(\boldsymbol{h})} \longrightarrow \mathbb{R}$ outputs an attention weight.

$$\omega_{i,j} := \mathrm{att}(\boldsymbol{W}\boldsymbol{h}_i, \boldsymbol{W}\boldsymbol{h}_j)$$

for an edge $(i, j)$ given the incident node embeddings $\boldsymbol{h}_i, \boldsymbol{h}_j$ to indicate the importance of the features of node $j$ to node $i$ for all node pairs $i, j \in \mathcal{V}$. Considering the neighborhoods given in the graph, the attention mechanism can be defined by

$$a_{i,j} := \mathrm{softmax}_j(\omega_{i,j}) = \frac{\exp \omega_{i,j}}{\sum_{k \in \mathcal{N}(i)} \exp \omega_{i,k}}.$$

### 3.2.3 Spatial and Spectral Graph Convolutions

Compared to the attention approach, the graph convolution aggregates the neighbored nodes directly using fixed weights (Bronstein et al., 2021) by

$$\boldsymbol{h}_u = \phi\left(\boldsymbol{x}_u, \bigoplus_{v \in \mathcal{N}(u)} c_{u,v}\psi(\boldsymbol{x}_v)\right), \tag{4}$$

where $c_{u,v}$ is a factor indicating the impact of neighbor $v$ on the hidden representation of node $u$. Note that $c_{u,v}$ is a pre-defined constant instead of a node-specific function, as is the case for attention layers. For spatial convolution, $c_{u,v}$ is usually given by the (weighted) adjacency matrix and thus includes structural information. For spectral convolution, spectral filters dependent on the graph Laplacian (c.f. 3.7.2) determine the weights of all nodes in the graph integrating structural information implicitly.

If the aggregation is a sum, the layer can be interpreted as linear diffusion or position-dependent linear filtering. An example of a spatial convolution in layer $l - 1$ is given in, e.g., Morris et al. (2019):

$$h_u^{(l+1)} = \sigma(W_1 h_u + W_2 \sum_{v \in \mathcal{N}(u)} h_v^{(l)}). \tag{5}$$

$W_1$ and $W_2$ are learnable weight matrices and $\sigma$ is an activation function such as $\mathrm{ReLU}(\cdot) = \max(0, \cdot)$ . An implementation of a standard spectral convolution is given in, e.g., Kipf & Welling (2017). The layer-wise propagation rule in layer $l$ is determined by

$$\boldsymbol{H}^{(l+1)} = \sigma(\underbrace{\tilde{\boldsymbol{D}}^{-\frac{1}{2}}\tilde{\boldsymbol{A}}\tilde{\boldsymbol{D}}^{-\frac{1}{2}}}_{\boldsymbol{L}_{norm}} \boldsymbol{H}^{(l)}\boldsymbol{W}^{(l)}), \tag{6}$$

where $\tilde{A} = A + E$ is the adjacency matrix with added self connections, $E$ is the identity matrix, $\tilde{D}$ is the degree matrix of the graph with self-loops, and $L_{norm}$ is the normalized graph Laplacian. $\sigma$ is an activation function, and $W$ is a learnable weight matrix functioning as a filter. The idea of spectral graph convolutions comes from signal processing. Vividly, one can imagine it as global message passing on a graph, weighted with a filter.

### 3.2.4 Recurrent Neural Networks

For each time step $t$, the Recurrent Neural Network (RNN) calculates a hidden representation using historical information together with the current input $X^{(t)}$ (Bronstein et al., 2021). First, the input is transformed by an encoder function $f$ to a representation vector $z^{(t)} = f(X^{(t)})$. Then, $z^{(t)}$ is aggregated together with the previous information by an update function $R : \mathbb{R}^k \times \mathbb{R}^m \to \mathbb{R}^m$ that additionally considers the hidden representation from the time step before. Altogether, a basic RNN is formalized by

$$h^{(t)} = R(z^{(t)}, h^{(t-1)}).$$

In the context of graph learning, the node feature matrix is commonly used as initial input $X^{(0)}$. Furthermore, in various GNNs for dynamic graphs, GCN layers are combined with RNNs by, e.g., modeling the GCN weight evolution with an RNN or propagating the learned structural information from one timestamp to the next timestamp.

## 4 Models Focusing on Structural Graph Properties

Learning on simple graphs is most prevalent in the research of GNNs. The elementary graph structure can already model relations in the data, and the mathematical foundations go back to the 17th century. After the most prominent introduction of Graph Neural Networks by Scarselli et al. (2009) for learning on static node-attributed graphs, many different extensions have been proposed. An overview of GNN models for simple static graphs is discussed in Sec. 4.1. Their approaches often build on the graph information processing scheme of Scarselli et al. and adapt it to new applications and several structural graph properties.

One of the extensions includes the higher-order representation of relational data with the aid of elementary hypergraphs. Hypergraph theory is still a young field and has been essentially developed by Claude Berge Berge (1973) in the 1970s. Learning on hypergraphs has also emerged as part of research in recent years and has much potential for applications on differently structured hypergraphs, as illustrated in Table 2.

### 4.1 GNNs for Simple Graphs

The number of GNNs for simple graphs has increased immensely in the past years, so Table 1 does not list all of them but gives an overview of several GNNs for simple graphs with structural properties defined in Def. 3.2. This demonstrates which graph types have already been considered in GNN research. The models are selected according to their up-to-dateness, relevance, general applicability, explicit addressing of a specific graph type, and simplicity, as discussed in the introduction. Note that in the case of processing attributed graphs, the attributes have to be encoded in $d$-dimensional vectors. To apply the corresponding models to arbitrary attributed graphs, preprocessing steps have to be utilized as in Zhang et al. (2019a).

The most common type of GNNs for simple graphs concerning structural properties are **convolutional** models, which compute new node representations in each layer. A common graph convolution as, for example, defined in GNN$^\star$ (Scarselli et al., 2009), or WL (Morris et al., 2019), typically assumes attributed nodes and allows for directed and undirected edges without being explicitly designed for either property. Models designed for other graph types typically extend a common spectral or spatial convolution to adapt to the specific structural property they focus on.

To consider directed edges, for example, in GenRecN (Sperduti & Starita, 1997), a standard spectral convolution is applied only on the out-neighbors of a target node, i.e., on those neighbors connected via a directed edge originating from it. A more recent model, MagNet (Zhang et al., 2021), applies a spectral convolution using a complex-valued Hermitian matrix, called magnetic Laplacian, instead of a symmetric and

real-valued Laplacian which cannot be computed due to the asymmetric adjacency matrix of a directed graph. The magnitude of the complex Laplacian indicates the presence of an edge but not its direction, while the phase of the complex Laplacian indicates the direction of an edge. The lack of models designed for graphs with edge attributes probably results from considering edge attributes only in addition to node attributes since edge attributes are typically not relevant in isolation. In terms of heterogeneity, a similar observation can be made. Corresponding graphs are either heterogeneous in their nodes and edges or node homogeneous, i.e. the, node set remains of one type. Edge heterogeneity is more common than node heterogeneity since it includes widely-used multi-relational graphs. These can be handled by, e.g., an extension of a standard convolution applied separately for each relation (GRNN (Ioannidis et al., 2019)).

Another common procedure is extending a convolutional model using an **attention** mechanism, e.g., as described in Sec. 3.2.2. Attention mechanisms are suitable for node-attributed graphs since they allow the computation of node-specific attention scores that express the importance of one node to another. These attention scores can serve as weights in computing node features to focus on specific nodes. Spectral (CapsGNN (Xinyi & Chen, 2018)), as well as spatial convolutions (GAT (Velickovic et al., 2018)), can be equipped with attention mechanisms. They can also be adapted to attributed edges to process entirely attributed graphs (EGNN (Gong & Cheng, 2019)). Also, attention-based models are a suitable approach for heterogeneous graphs since multi-head attention can be used to model different relation types, as in AA-HGNN (Ren et al., 2020). A particular case of attention convolution is HAN (Wang et al., 2019), which utilizes a selected set of meta paths for neighborhood aggregation.

Table 1: **GNN's developed for learning on simple graphs of different structures.** Such models are most prevalent in the research of GNNs.

| | Graph Type | Models | Problem | Data Category |
|---|---|---|---|---|
| graph | directed | GenRecN Sperduti & Starita (1997) | graph classification | logic terms |
| | | MagNet Zhang et al. (2021) | link prediction, node classification | linking of websites, synthetic |
| | undirected | GNN* Scarselli et al. (2009) | subgraph matching, graph classification, web page ranking | synthetic, mutagenesis (molecules) |
| | node-attributed | GAT Velickovic et al. (2018) | node and graph classification | citation networks, protein interaction |
| | | CapsGNN Xinyi & Chen (2018) | | biological-, social networks |
| | | WL Morris et al. (2019) | graph classification, attribute prediction | biological-, social networks, molecules |
| | edge-attributed | — | | |
| | attributed | EGNN Gong & Cheng (2019), PG-GNN Xia & Ku (2021) | graph classification, node and edge attribute prediction | citation networks, protein structure |
| | node-heterogeneous | — | | |
| | edge-heterogeneous | GRNN Ioannidis et al. (2019) | node classification | citation networks |
| | heterogeneous | AA-HGNN Ren et al. (2020), HAN Wang et al. (2019) | | news articles & citation networks |
| | multi | — | | |

## 4.2 GNNs for Hypergraphs

Learning on hypergraphs as in Def. 3.1 has been rarely explored. Most approaches involve convolutions adapted to hypergraphs, i.e., the property that an edge can be incident to an arbitrary number of nodes. Table 2 lists GNNs that mainly address node classification on citation networks represented as hypergraphs, which shows that the application of hypergraphs is not yet widespread; hence, the available datasets are currently limited. During the research for hypergraph GNNs, it turned out that, so far, only a few GNNs have been applied to hypergraphs. When it comes to additional structural properties as defined in Def. 3.2, sometimes only one or two models for the specific hypergraphs have been developed. Table 2 indicates that the data is still very homogeneous and that the heterogeneity in graphs is only addressed to a limited extent.

One option to handle hypergraphs is to **transform them into simple graphs** and apply standard GNNs afterward. This preprocessing can be done by selecting two representative nodes for each hyperedge, as in HyperGCN (Yadati et al., 2019). Based on the assumption that nodes in a hyperedge are similar, the representative nodes typically have the most significant difference between their attributes. Another approach presented in LHCN (Bandyopadhyay et al., 2020) represents a hypergraph as a line graph[6]. In this process, each hyperedge of the original graph serves as a simple node in the line graph. The corresponding node attributes are computed by the average across the attributes of all hypernodes in that hyperedge. Both variants allow for processing attributed and undirected hypergraphs using GNN models for simple graphs.

Table 2: **GNNs learning on hypergraphs with different additional properties.** The selection of GNNs is still limited, which illustrates gaps and the potential of the young research field.

| | Graph Type | Models | Problem | Data Category |
|---|---|---|---|---|
| hypergraph | directed | NDHGNN
Tran & Tran (2020) | node classification | citation networks |
| | undirected | HyperGCN
Yadati et al. (2019) | densest k-subhypergraph problem, node classification | combinatorial optimization, citation networks |
| | | HyperConvAtt
Bai et al. (2021) | node classification | citation networks |
| | node-attributed | LHCN
Bandyopadhyay et al. (2020) | node classification | citation networks |
| | edge-attributed | HGNN
Feng et al. (2019) | node classification, object classification | citation networks |
| | attributed | AHGAE
Hu et al. (2021) | graph clustering | citation networks, 3D models |
| | node-heterogeneous | — | | |
| | edge-heterogeneous | — | | |
| | heterogeneous | HWNN
Sun et al. (2021b) | node classification | citation networks |
| | multi | G-MPNN (multiple edges)
Yadati (2020) | link prediction | knowledge (hyper-) graphs |

There are also models which are **specifically designed for hypergraphs**, most of them based on spectral convolutions. The graph's incidence matrix can be used to adapt spectral convolutions to attributed

---

[6]The nodes of a line graph w.r.t. an original graph are determined as the edges of the original graph, while the edges are inserted between two edges of the original graph that share an incident node.

hypergraphs and the node and edge degree matrix in the neighborhood aggregation (HGNN (Feng et al., 2019), AHGAE (Hu et al., 2021)). Such a convolution can be additionally equipped with an attention mechanism (HyperConvAtt (Bai et al., 2021)). NDHGNN (Tran & Tran, 2020) uses separate incidence matrices for the source and target nodes to model the graph's Laplacian in the spectral hypergraph convolution to process directed hypergraphs. Such convolutions can also be used for heterogeneous graphs by using edge homogenization, e.g., by working on subgraphs that include hyperedges of only one specific type. In HWNN (Sun et al., 2021b), the spectral convolution is applied on subgraphs, which include hyperedges of only one specific type. Finally, to enable learning on multi-hypergraphs, a message-passing GNN is extended to include multiple relations and node or edge duplicates (Yadati, 2020).

Both approaches, i.e., transforming hypergraphs into simple graphs or directly working on them, have advantages and disadvantages. The first case enables the application of well-established GNN architectures, which have typically been investigated more thoroughly. However, the transformation is often related to information loss, affecting performance. In HyperGCN (Yadati et al., 2019), the information from nodes in hyperedges that do not serve as representative nodes disappear. Models that directly operate on hypergraphs, such as HGNN (Feng et al., 2019), can use the complete information to learn.

## 5 Models Respecting Dynamic Graph Properties

Many applications include data that changes with time. In the application of graphs, it often appears, e.g., that graphs are growing or structurally changing or that node and edge attributes are evolving, as defined in Def. 3.3. Therefore, the research on GNNs for dynamic graphs has expanded immensely. There are two common approaches in graph learning for representing a graph's dynamical behavior: discrete-time and continuous-time representation (cf. Def. 3.4). The first approach has been widely used since the snapshots simplify the processing of structures in the graph. Corresponding proposed GNNs in the literature for processing discrete-time graphs are listed in the next section. The continuous-time approach is much more compact in its representation since it stores only events instead of the entire graph for each time step. However, a local evaluation of the graph is required, i.e., the area in the graph affected by an event has to be identified and retrieved to process the event using a GNN. Hence, the application of this representation is more complex, which is also reflected in the less frequent use of it in GNN models, which can be seen in Sec. 5.2.

### 5.1 GNNs for Discrete-Time Graphs

Although dynamic graphs are much more challenging to handle than static graphs due to the additional temporal dependencies, existing dynamic GNN models already cover many structural graph properties. GNN models operating on discrete-time graphs are typically extensions of static GNNs since the discrete-time representation corresponds to a series of static graphs. Therefore, similar gaps appear, i.e., only node-heterogeneous graphs and multi-graphs are not yet covered. The structural component of dynamic graphs can be learned by applying standard GNN models to the graph snapshots.

Those models are often combined with **RNN-based** models to encode the dynamics, which capture the temporal features. Such an approach is pursued in, e.g., GCRN (Seo et al., 2018). The model processes attribute dynamic graphs using a spectral GCN combined with an LSTM. First, the node attributes are preprocessed by a spectral convolution, and the resulting representation is passed to the LSTM, which captures the data distribution. A similar approach is taken in WD/CD-GCN (Manessi et al., 2020), which applies a GCN to transform the input graph sequence into a sequence of node representations, which are then processed by a modified LSTM. EpiGNN (La Gatta et al., 2020) also combines GCNs and LSTMs to predict the parameters of a generic epidemiological model based on historical movement data. The model embeds the graph nodes for each time step, representing locations using a standard GCN. It learns the desired parameters by embedding the current graph and information from previous time steps stored in the LSTM.

Table 3: **GNN's learning on discrete-time dynamic graphs.** Many of these models are extensions of the static case since the discrete-time representation corresponds to a series of static graphs. Therefore also the gaps appear similar to the static case. The □ sign means, that the graph handled by the model can have attributes, but the attributes are static. Thus, they appear or disappear together with their respective nodes or edges but do not change over time.

| Graph Type | | | Models | Nodes | | Edges | | Attr. | | Problem | Data Category |
|---|---|---|---|---|---|---|---|---|---|---|---|
| | | | | add | del | add | del | node | edge | | |
| DTR | graph | directed | EpiGNN La Gatta et al. (2020) | × | × | × | × | ✓ | □[7] | node label prediction | covid-19 |
| | | undirected | DySAT Sankar et al. (2020) | × | × | ✓ | ✓ | × | □ | link prediction | communication, rating networks |
| | | | (WD/CD)-GCN Manessi et al. (2020) | × | × | ✓ | ✓ | ✓ | × | node classification | research community |
| | | node-attributed | GCRN Seo et al. (2018) | × | × | × | × | ✓ | □ | video prediction,[8] graph sequence prediction | videos, text |
| | | edge-attributed | DynGEM[9] Goyal et al. (2018) | ✓ | ✓ | ✓ | ✓ | × | □ | graph reconstruction, link prediction, anomaly detection | synthetic, collaboration, communication networks |
| | | attributed | EvolveGCN Pareja et al. (2020) | ✓ | ✓ | ✓ | ✓ | ✓ | □[10] | link prediction, edge and node classification | synthetic, social networks, bitcoin, community network |
| | | node-heterogeneous | — | | | | | | | | |
| | | edge-heterogeneous | RE-Net Jin et al. (2019) | × | × | ✓ | ✓ | × | × | extrapolation link prediction | knowledge graphs |
| | | heterogeneous | DyHAN Yang et al. (2020) | ✓ | ✓ | ✓ | ✓ | × | × | link prediction | e-commerce, online-community |
| | | multi | — | | | | | | | | |

[7]Only static edge attributes are considered.
[8]The model uses the moving written digits datasat (moving-MNIST dataset) generated by Shi et al. Shi et al. (2015)
[9]Only considers the previous time step, patterns of short duration (length 2) for link prediction and is restricted to weights.
[10]Edges are weighted not generally attributed.

| | | | | | | | | | task | data |
|---|---|---|---|---|---|---|---|---|---|---|
| hypergraph | directed | DHAT Luo et al. (2021) | × | × | × | × | ✓ | × | feature prediction | traffic data |
| | undirected | — | | | | | | | | |
| | node-attributed | STHAN-SR Sawhney et al. (2021) | × | × | × | × | ✓ | □ | node ranking | stock prediction |
| | | HGC-RNN Yi & Park (2020) | × | × | × | × | ✓ | × | feature prediction | traffic flows |
| | attributed | Hyper-GNN Hao et al. (2021) | × | × | ✓ | ✓ | ✓ | □[10] | action recognition/graph classification | human motion |
| | node-, edge heterogeneous | — | | | | | | | | |
| | heterogeneous | MGH Yan et al. (2020) | ✓ | ✓ | ✓ | ✓ | ✓ | ✓ | graph classification | videos |
| | multi | — | | | | | | | | |

Further approaches combining RNNs and GCNs are, e.g., RE-Net (Jin et al., 2019) and EvolveGCN (Pareja et al., 2020). RE-Net computes local representations for all nodes by applying an RNN to its temporal neighborhood, i.e., the neighbors at different time steps. The model is designed for edge-heterogeneous graphs and aggregates the edges of different types using a GCN before the neighborhood aggregation. To obtain a global node representation, the local node representations over time are processed by another RNN. In contrast to the models mentioned above, EvolveGCN uses the RNN to model the GCN weights, which embeds the graph nodes. More specifically, the weights of each GCN layer are generated by an RNN, which takes the weights of the preceding GCN layer and, optionally, the node embeddings as input. This way, the model adapts the GCN weights along the temporal dimension to tackle the problem of changing node attributes.

Another way to handle temporal features in GNNs is **temporal attention**. DySat (Sankar et al., 2020), e.g., generalizes the GAT approach (Velickovic et al., 2018) described in Sec. 1 to dynamic graphs. On the one hand, the model extends the structural attention mechanism. On the other hand, it incorporates an additional temporal attention mechanism that enforces an auto-regressive behavior. Similarly, DyHAN (Yang et al., 2020) generates node embeddings using node-level attention and updates them via neighborhood aggregation and edge-level attention. Finally, the node embeddings are aggregated over time using a temporal attention mechanism. Heterogeneity is accounted for by applying node-level attention at each time step to subgraphs of only one edge type. During edge-level attention, the importance of each edge type is learned through a one-layer MLP. DynGEM (Goyal et al., 2018) takes an entirely different approach and is a dynamically extendable autoencoder for growing graphs. The input and output dimensions are extended respectively for each new incoming node.

Since handling hypergraphs is challenging, especially in the dynamic case, few models have been proposed yet. One model that combines RNNs and GCNs is the HGC-RNN (Yi & Park, 2020). It integrates the temporal evolution of higher-ordered structures with two different hypergraph convolutions to encode structural and temporal information, global states, and a subsequent recurrent unit. All other hypergraph models in Tab. 4 involve attention mechanisms. Typically, they combine a hypergraph convolution with a temporal attention mechanism, as in DHAT (Luo et al., 2021), and MGH (Yan et al., 2020). For graph learning on video data, MGH extracts features from classical CNNs of different granularity to define hypergraphs of different types beforehand. The heterogeneity of the edges is then integrated into the model using corresponding attention.

A very similar model is Hyper-GNN (Hao et al., 2021). It applies a hypergraph convolution similar to HGNN (Feng et al., 2019) from Sec. 4.2 and a corresponding attention mechanism adapted to neighborhoods

on hypergraphs. The overall architecture consists of three parallel networks of the same structure, each processing different input features. STHAN-SR (Sawhney et al., 2021) also applies an attention convolution, which has been designed for static graphs. It applies a HyperGCN model (Yadati et al., 2019) from Sec. 4.2 to process node features that have been generated utilizing an LSTM and a temporal attention mechanism.

When considering the types of dynamics the different models can handle, it becomes apparent that most of them focus on specific dynamics, such as dynamic node attributes only (EpiGNN (La Gatta et al., 2020), GCRN (Seo et al., 2018), STHAN-SR (Sawhney et al., 2021), HGC-RNN (Yi & Park, 2020), DHAT (Luo et al., 2021)) or evolving edges (RE-Net (Jin et al., 2019), Hyper-GNN (Hao et al., 2021), WD/CD-GCN (Manessi et al., 2020)). In particular, to the best of our knowledge, MGH (Yan et al., 2020) is the only model capable of processing graphs with changing node and edge sets and node and edge attributes over time. Among all the dynamics, deleting nodes and changing edge attributes have emerged as the least considered and probably most challenging ones. In the case of decreasing node sets, difficulties arise from the changing data structures leading to data gaps, the handling of obsolete data, and in particular, the lack of data and applications in this area. At the same time, the lack of models for changing edge attributes is a consequence of the fact that there are hardly any data and applications for this case.

## 5.2 GNNs for Graphs in Continuous Time

Regarding dynamic graphs in continuous-time representation, fewer models use the advantages of this compressed representation, although the amount is growing quickly. Especially dynamic hypergraphs in this form are currently rarely investigated. Using the continuous-time representation allows the usage of explicit timestamps and an explicit specification of the change in the graph instead of processing a graph snapshot in every time step. Therefore, it drastically reduces the storage requirements. Nevertheless, utilizing this representation is challenging due to the absence of a direct encoding of the graph structure at a particular time and the model's requirement to be updateable in case of occurring events.

**Stochastic processes** are frequently used to model dynamic graphs represented as a sequence of events. Typically, such processes model the probability of an event occurring at a specific time. These events encode the graph's dynamics, such as a node's appearance or an attribute's change, and an intensity function describes the distribution of the events over time. The occurrence of an event is modeled based on the most recent events involving the nodes or edges of interest.

Examples of approaches utilizing stochastic processes are Know-Evolve (Trivedi et al., 2017) and its extension, DyREP (Trivedi et al., 2019). Know-Evolve considers events of appearing edges of different types. Here, separate embeddings for source and target nodes are computed to take directed edges into account. Furthermore, a learnable function is applied to the difference between a specific node's current and the last event to capture the temporal evolution. Moreover, previous embeddings of the nodes and edges involved in the current event are processed by an RNN-based model to encode the effect of the recurrent participation of each entity in events. The node embedding is further processed by a learnable function, which captures the compatibility of nodes in previous edges. Based on the learned node embeddings, a temporal point process is used to model the probability of an edge occurring between two existing nodes at the next timestamp. Know-Evolve's extension DyREP additionally uses structural information of the graph for two different edge types that represent different ways of communication between nodes.

A different approach is proposed in DyGNN (Ma et al., 2020). It utilizes **LSTMs** in two kinds of units, one for the source and the other for target nodes connected through an edge. In the case of link prediction, node pairs are ranked respecting the cosine similarity of their node representations, and in the case of node classification, the softmax function is utilized. Similarly, TGN (Rossi et al., 2020) enables the usage of a memory module, which can be updated using an RNN such as an LSTM or GRU. The obtained node embedding can be used together with a learnable function to perform, e.g., temporal attention, summation, or projection. Afterward, an MLP processes the node embeddings of node pairs to generate a probability for the edge at the next timestamp to perform future link prediction.

(Souza et al., 2022) proved a deficit in the expressivity of TGN and proposed the Positional-Encoding Injective Temporal Graph Net (PINT) to overcome this deficit. A node embedding at a certain timestamp is determined by an injective temporal aggregation of the neighborhood, where the attributes of the neighbors are calculated

with an MLP over the neighboring node attribute concatenated with the corresponding edge attribute. Further, the performance is improved by concatenating the relative positional features that incorporate information from the temporal walk structures with the node features.

Jin et al. (2022) propose a model that uses a recurrent unit combined with temporal walks. Such walks are defined as node sequences in which subsequent nodes have previously been involved in an event together. While sampling temporal walks for a target node, nodes that have been involved in an event with the target node more recently are sampled with a higher probability. By anonymizing these walks into relative and node-unspecific encodings, so-called motifs are created. An autoregressive gated recurrent unit processes these to compute the node embeddings. Since the nodes are sampled irregularly for the temporal walks, the motifs are integrated over multiple interaction time intervals.

Table 4: **GNN's learning on continuous-time dynamic graphs.** Due to the difficulties arising from the lack of a direct encoding of the graph structure at each time point, there are only certain graph types covered by models models utilizing graphs in this representation.

| Graph Type | | | Models | Nodes | | Edges | | Attr. | | Problem | Data Category |
|---|---|---|---|---|---|---|---|---|---|---|---|
| | | | | add | del | add | del | node | edge | | |
| CTR | graph | directed | Know-Evolve Trivedi et al. (2017) | × | × | ✓ | × | × | × | link/time prediction | socio-political interactions |
| | | | DyGNN Ma et al. (2020)[11] | ✓ | × | ✓ | × | × | × | link prediction, node classification | communication/ trust networks |
| | | undirected node-attributed | DyRep Trivedi et al. (2019) | ✓ | × | ✓ | × | × | × | link/event time prediction | social networks, github |
| | | edge-attributed | — | | | | | | | | |
| | | attributed | NeurTWs Jin et al. (2022) | × | × | ✓ | × | × | × | dynamic link prediction | social networks, interaction networks |
| | | node-heterogeneous | — | | | | | | | | |
| | | edge-heterogeneous | Know-Evolve Trivedi et al. (2017) | × | × | ✓ | × | × | × | link/time prediction | socio-political interactions |
| | | | DyRep Trivedi et al. (2019) | ✓ | × | ✓ | × | × | × | link/event time prediction | social networks, github |
| | | heterogeneous | — | | | | | | | | |
| | | multi | PINT Souza et al. (2022), TGN Rossi et al. (2020) | ✓ | ✓ | ✓ | ✓ | ✓ | ✓ | node classification, edge prediction | social networks |

[11]The baseline models used in the experiments are made for continuous-time dynamic GNNs. All the models used are either made for static graphs (e.g., GCN, GraphSAGE) or discrete-time dynamics (e.g., DynGEM, DANE, Dynamic Triad).

| | | | | | | | | | | | |
|---|---|---|---|---|---|---|---|---|---|---|---|
| hypergraph | directed | — | | | | | | | | | |
| | undirected | HIT Liu et al. (2021) | □[12] | × | ✓ | × | × | × | edge-, pattern-, time prediction | Q&A platform, political interactions, patient medication |
| | (node, edge) attributed, (node, edge) heterogeneous, multi | — | | | | | | | | | |

To the best of our knowledge, the only GNN developed for hypergraphs in continuous-time representation is HIT (Liu et al., 2021). To encode structural and temporal information, it uses temporal random walks defined as a randomly selected set of hyperedges backward in time. Afterward, an aggregation mechanism pools the obtained representation into the final node embedding.

## 6 Models Utilizing Semantic Graph Properties

Besides the structural graph properties, it is also possible to consider semantic properties in designing a GNN. Although semantic graph properties typically do not explicitly affect the graph's structure (R-5), it can be advantageous to leverage them in GNNs since the graph topology could change or specialized architectures might better preserve the original properties in the learned representation. The necessity for this comes from the data's nature and theoretical considerations to learn structures more efficiently or to explicitly model certain constraints or properties of the data structure. The semantic properties listed in Def. 3.2 are a selection from data-motivated characteristics (e.g., bipartite nodes for user-item modeling, complete graphs for relation prediction) and graph theory (e.g., regular or disconnected graphs, trees). Accordingly, GNNs that integrate some of these properties are presented in Tab. 5.

Some characteristics are considered more often in graph learning than others. These include, e.g., complete, acyclic, and bipartite graphs since they reflect frequently occurring characteristics of graph applications. In contrast, recursive graphs or (poly-)trees are considered explicitly only occasionally. To the best of our knowledge, regular graphs do not play a significant role in graph learning.

**Complete graphs** represent the existence of a connection between each node pair. Standard Message-Passing GNNs or Convolutional GNNs are theoretically capable of handling complete graphs. However, especially in the case of GCNs, the neighborhood of all nodes is considered equally, and thus, the information flow in the graph is inexpressive. Some GNNs have been developed for complete graphs to overcome this problem.

MGCN (Lu et al., 2019), e.g., is specifically designed to predict properties of molecules represented as a complete graph. The network is a standard Message-Passing Neural Network (MPNN), utilizing node and edge attributes. The crucial innovation is how nodes and edges are embedded. The idea is to model quantum interactions between atoms since these influence the overall properties of the molecule. Initially, the atoms of a molecule define nodes of a complete graph, respecting the number of protons in their nuclei. Then, the edge attributes are constructed using a radial basis function (RBF) layer and processed in a hierarchical GCN to weight nodes in the message-passing. A final node embedding is obtained by executing several convolutions and hierarchically aggregating the neighborhood of increasing depth. The learned node and edge embeddings are summed across the graph to infer the prediction of molecule properties. Since molecule datasets typically comprise labels only for a small fraction of the data and only for smaller molecules, the authors mainly focused on generalizability and transferability between different molecule sizes.

As can be observed from the table, there are models focusing on **bipartite graphs**, i.e., graphs that can be divided into two disjoint node sets such that every edge connects a node of one set to a node from the other

---

[12]Nodes only appear together with new hyperedges.

set. One example is BGNN (He et al., 2019), which focuses on generating a suitable representation for such graphs. For this purpose, information across and within the graph's two partitions (domains) is aggregated to enable inter-domain message passing. The model is trained in a so-called cascade way, i.e., the training of a layer begins after the preceding layer has been fully trained. Thereby, the loss function for the domains is defined layer-wise. Together with a global loss, the quality of the resulting node representation is measured.

BipGNN (Wang et al., 2020), in contrast, restricts the convolution to the propagation between the disjoint node sets. The network encoder produces pairwise embeddings for nodes from the two disjoint sets, and the decoder maps these embeddings to an association matrix to perform link prediction between disjoint sets.

In the case of **unconnected graphs**, the underlying concept of information flow in GNNs may reach its limits. In standard GNNs, subgraphs without a connection to the rest of the graph are processed isolatedly. Thus, small isolated subgraphs may not provide enough structural information to prevent over-smoothing (Li et al. (2018) of the GNN. To tackle this problem, the generative graph transformer network (GTN) (Yun et al., 2019), e.g., aims to identify valuable connections between unconnected nodes to the rest of the graph. It enables learning on multiple subgraph structures in a heterogeneous graph by concatenating graph convolutions on different meta-paths.

Table 5: **GNNs using semantic graph properties.** The specific semantic properties have been selected due to their rather common appearance in graph data. Since some graph characteristics are considered in more applications or in more popular applications, there exist more GNN models, e.g. bipartite graphs are considered often.

| Graph Type | Models | Problem | Data Category |
|---|---|---|---|
| complete | MGCN Lu et al. (2019) | graph attribute prediction | quantum chemistry |
| $r$-regular | — | | |
| bipartite | BipGNN Wang et al. (2020) | link-rank prediction | drug repurposing |
| | BGNN He et al. (2019) | node representation learning | social-/citation networks |
| unconnected[13] | GTN Yun et al. (2019) | graph generation, meta-path generation, node classification | citation networks, movie genres |
| acyclic[14] | DAGNN Thost & Chen (2021) | node prediction, longest path prediction | source code, neural architectures, Bayesian networks |
| trees | GenRecN Sperduti & Starita (1997) | graph classification | logic terms |
| polytrees | CTNN He et al. (2021) | node classification | 3d surfaces in context of hydrological applications |
| recursive | MPNN-R Yadati (2020) | node classification | documents in academic networks |
| hierarchical | Hier-GNN Chen et al. (2022) | image classification | images |

**Acyclic graphs** occur across various domains, such as source code, neural architectures, or logic terms. DAGNN (Thost & Chen, 2021) learns a representation for directed acyclic graphs driven by the partial order

---

[13]Since, to the best of our knowledge, most of the models do not specify the connectedness, it is assumed here that they can handle both connected and unconnected graphs.

[14]The majority of GNN models in this survey can handle cycles because they are very common in graphs.

over the graph nodes. It is an RNN-based message-passing network utilizing an attention module to obtain the messages, which are then forwarded through a GRU. A graph representation is obtained by first concatenating the source and target node representations separately, then max-pooling them and concatenating the results.

Particular cases of acyclic graphs are **trees**, i.a., examined in GenRecN (Sperduti & Starita, 1997). This early work, as mentioned in Sec. 4.1, applies a spatial neighborhood convolution on the out-neighbors of a node. Polytrees also serve as suitable representations for some types of data, such as surface contours of 3D data. As shown in CTNN (He et al., 2021), polytrees can be used to model the evolution of the surface contours at different elevation levels. The model uses a U-Net (Ronneberger et al., 2015) architecture with ChebyNet (Defferrard et al., 2016) and diffusion Graph Convolution (Li et al., 2017) Layers, using graph pooling and unpooling methods for the characteristic unit architecture.

Another model using a U-Net (Ronneberger et al., 2015) architecture is Hier-GNN (Chen et al., 2022), which explores hierarchical correlations between nodes. For this purpose, specialized pooling and unpooling methods are explicitly defined to encode hierarchical information. Graph convolutions are then applied among a layer in the hierarchy.

Finally, MPNN-R (Yadati, 2020) has been developed to encode **recursive graphs**. It is based on G-MPNN (Yadati, 2020) mentioned in Sec. 4.2 and adapts the message-passing function for recursive multi-relational ordered multi-hypergraphs.

## 7 Models for Combined Graphs

Arbitrary combinations of graph types can be used to model real-world problems and thus be considered for graph learning purposes. To conclude this work, we give a selected list of graph-type combinations used in several research fields where GNNs are already established. Therefore, this list is not necessarily complete but gives an insight into further research on GNNs considering combined graph types.

The architectures listed in Tab. 6 are GNN models specialized for a particular combination of graph properties. Some of them use a selected non-Euclidean space that is assumed to provide a better fit for the specific data. GCN (Kipf & Welling, 2017), e.g., defines a standard spectral graph convolution for simple graphs allowing for one-dimensional edge weights, whereas Hyperbolic GNN (Liu et al., 2019) defines its extension to hyperbolic space. Hyperbolic GNN operates on Riemannian manifolds[15] and is independent of the underlying space. Since every point of a differentiable Riemannian manifold can be approximated by Euclidean space, all functions with trainable parameters are executed in Euclidean Space. HVGNN (Sun et al., 2021a) uses a hyperbolic model as well. More precisely, it consists of a temporal graph neural network based on convolution and attention modules and a variational graph auto-encoder in hyperbolic space. A map from time to a hyperbolic space encodes time information to handle time in the convolution process. This way, the aggregation of the features is done in a time-aware neighborhood.

The combined graph structures that make up **knowledge graphs** represent a common application for GNNs. They can represent all types of attributes, together with heterogeneity and dynamics. Therefore, many different models have been developed. KGIN (Wang et al., 2021), e.g., uses an attention mechanism that combines different relations into so-called intents to model the user-item relations. These are subsequently used for user and item embeddings modeled via another attention layer to predict the probability of a user adopting an item. A similar use case is approached in SBGNN (Huang et al., 2021), where the two node types of users and items are represented as a bipartite graph connected through signed relations. The model uses a message-passing scheme, including an attention mechanism to encode positive and negative links in recommender, voting, and review systems. HetG (Zhang et al., 2019a) processes a similar graph type. The model is designed to embed heterogeneous graphs with node and edge attributes of any kind. It generates a heterogeneous neighborhood using Random Walk with Restart and applies a Bi-LSTM for heterogeneous content embedding. Different types of nodes are then combined via an attention mechanism.

---

[15]A Riemannian manifold is a real and smooth manifold equipped with an inner product at each point of the manifold (Liu et al., 2019).

A particular type of graph that can be useful for several applications is the **multiplex graph**. It consists of different layers, each with the same set of nodes but different sets of edges within these layers. Inter-layer edges connect the same nodes across different layers. In MXMNet (Zhang et al., 2020a), a two-layer multiplex graph is utilized such that the so-called local layer is generated with the aid of molecular expert knowledge, and the global layer depends on the neighborhood of the local layer. MXMNet applies a message-passing procedure to each layer separately and enables communication between these layers by defining an additional cross-layer mapping function.

Multiplex graphs can also be used to model temporal features without explicitly using a dynamic graph representation as in STGNN (Kapoor et al., 2020). This model is specifically designed to predict the daily new cases of COVID-19 in a particular region based on mobility data. Each layer of the multiplex graph corresponds to a specific period, i.e., a day. Nodes represent regions, and relations within these layers describe human mobility between different regions. Edges between the layers are temporal and define a node's attribute through time. STGNN processes such graphs using spectral graph convolutions.

Table 6: **GNN's for combined graph types.** The graph type combinations were selected to cover combinations in fields where the usage of GNNs is already established.

| Graph Type | Models | Problem | Data Category |
|---|---|---|---|
| undirected node-attributed | GCN Kipf & Welling (2017) | node classification | citation networks, knowledge graphs |
| | Hyperbolic GNN Liu et al. (2019) | graph classification, node regression | synthetic, molecular, blockchain |
| knowledge graph | KGIN Wang et al. (2021) | link prediction | recommender systems |
| content-associated heterogeneous | HetG Zhang et al. (2019a) | link prediction, recommendation, (inductive) node classification, node clustering | review networks |
| multiplex | MXMNet Zhang et al. (2020a) | graph feature prediction | molecules |
| spatio-temporal | STGNN Kapoor et al. (2020) | node attribute prediction | disease spreading |
| multi-relational bipartite | SBGNN Huang et al. (2021) | link sign prediction | recommender, voting, review systems |
| bipartite edge-growing in continuous-time | JODIE Kumar et al. (2019) | future user-item interaction prediction, user state change prediction[16] | social media, wiki, music, student actions |
| undirected node-attributed edge-dynamic | HVGNN Sun et al. (2021a) | link prediction, node classification | social, citation, knowledge |

JODIE (Kumar et al., 2019) also processes temporal information, but it directly encodes the dynamics using an RNN. It can be considered a particular case of the TGN-Model (Rossi et al., 2020). For node embeddings, it also uses a memory module that can be updated using an RNN. The message-passing function is set to the identity and applied together with a learnable time projection function. The model is evaluated, e.g., on link prediction between users and items inferred from the distance between the embeddings of a pair of nodes.

---

[16]The task is to predict if an interaction will lead to a state change in user, particularly in two use cases: predicting if a user will be banned and predicting if a student will drop-out of a course.

## 8   Conclusion and Future Work

This survey provides a fine-grained overview of Graph Neural Networks for graph types of different structural constitutions. To the best of our knowledge, this is the first work to survey which graph types are addressed by published GNNs. We overviewed and defined the most common graph types and properties and the respective GNN models. Moreover, we identified GNN models specialized for specific graph properties and investigated how they handle these. This way, we could relate formal graph properties to the corresponding practical GNN models. Furthermore, we analyzed the architecture of the considered models and grouped them according to the modules they apply, i.e., the type of layer, such as convolutional or recurrent layers. Additionally, we analyzed GNN models concerning dynamics and grouped the models according to the types of dynamics they can process.

Our work allows several conclusions to be drawn and identifies gaps concerning the graph types, properties, and dynamics that GNN models can handle. First, existing GNN models can, in principle, handle the most common structural graph properties (e.g., attributed, directed, node-heterogeneity) for static graphs and hypergraphs. The lack of models for a few properties results from the existence of more general models, e.g., there is no GNN model for node-heterogeneous graphs in discrete time, but there is one for fully heterogeneous graphs. Another reason could be a lack of standard graph data sets for such types. Furthermore, there are many GNN models which consider graphs in discrete-time representation. These models cover the most common graph types and properties except for the multiplicity of nodes or edges. A difficulty in handling multiplicity results from the inability of a standard graph's adjacency matrix to encode duplicate nodes.

When it comes to the models for graphs in continuous time, it is evident that there are substantial gaps in research on GNNs for most of the graph types compared to the discrete case. In particular, only one model for hypergraphs has been found. Generally, developing GNNs for continuous-time graphs is still a young field of GNN research. Another reason for the small number of graph types covered by models for continuous-time graphs could be that such models typically use stochastic point processes to model the dynamic behavior of the graphs. The number of different events increases with the number of dynamic graph properties considered. Since a point process models each event, the model becomes more complex. From the results from Tab. 4, it can be observed that most events model discrete outputs in continuous time, such as whether there is a new edge. When including attribute dynamics for real-valued attributes, the model must deal with continuous values in continuous time, making the model more computationally intensive.

Most dynamic graphs addressed by GNN models exhibit only specific dynamics, such as strictly growing graphs or dynamic node attributes. GNN models for graphs with dynamics in the edge attributes and the deletion of nodes are scarce. To the best of our knowledge, only one model, MGH(Yan et al., 2020), has been developed to process graphs with all dynamics considered in this work, i.e., changing node and edge sets as well as changing node and edge attributes. Reasons for this may be the popularity of problems where graphs are growing over time and node deletions are believed not to play a crucial role (as, e.g., in citation networks, recommender systems, or data networks) and the difficulties that arise when combining known GNN techniques for dynamic graphs. Considering the discrete-time representation of graphs, e.g., GNN techniques for static graphs are usually applied to every graph snapshot and combined with an RNN to capture the dynamics, leading to computationally expensive models.

Finally, existing GNN models have been developed to cover many semantic graph properties or for particular combined graph types dependent on the given data structure, which shows that multiple graph properties can be learned simultaneously by GNN models.

To sum up, the research on GNNs for particular graph types has become a hot area in recent years. However, this extensive survey could reveal gaps in graph types, properties, and dynamics that are not yet considered sufficiently in the GNN community.

## 9 Notation

Table 7: Notation used throughout this work.

| | |
|---|---|
| $\mathbb{N}$ | natural numbers |
| $\mathbb{N}_0$ | natural numbers starting at 0 |
| $\mathbb{R}$ | real numbers |
| $\mathbb{R}^k$ | $\mathbb{R}$ vector space of dimension $k$ |
| $\lvert a \rvert$ | absolute value of a real $a$ |
| $\lVert \cdot \rVert$ | norm on $\mathbb{R}$ |
| $\lvert M \rvert$ | number of elements of a set $M$ |
| $\varnothing$ | empty set |
| $\{\cdot\}$ | set |
| $\{\!\!\{\cdot\}\!\!\}$ | multiset, i.e., set allowing multiple appearances of entries |
| $\cup$ | union of two (multi)sets |
| $\uplus$ | disjoint union of two (multi)sets |
| $\subseteq$ | sub(multi)set |
| $\times$ | factor set of two sets |
| $\psi$ | learnable message function |
| $\phi$ | activation function |
| $\sigma$ | sigmoid activation function |
| $\oplus$ | aggregation |
| $\parallel$ | concatenation |
| $\boldsymbol{A}$ | adjacency matrix |
| $\tilde{\boldsymbol{A}}$ | adjacency matrix with self-loops |
| $\boldsymbol{B}$ | edge degree matrix |
| $\boldsymbol{D}$ | node degree matrix |
| $\tilde{\boldsymbol{D}}$ | node degree matrix with self-loops |
| $\boldsymbol{E}$ | identity matrix |
| $\boldsymbol{I}$ | incidence matrix |
| $\boldsymbol{L}$ | Laplacian matrix |
| $\boldsymbol{W}$ | edge weight matrix |

**Author Contributions**

All authors contributed equally.

**Acknowledgments**

The GAIN project is funded by the Ministry of Education and Research Germany (BMBF), under the funding code 01IS20047A, according to the 'Policy for the funding of female junior researchers in Artificial Intelligence'.
The authors would like to thank Mohamed Hassouna, and Jan Schneegans for the fruitful discussion and corrections of the manuscript.

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

## 10  Appendix

### 10.1  Reasons for Selection of the Models per Graph Type

To clarify why each model was chosen for a certain graph type, the reasons are listed in Tab.8. Additionally, the criterion of generality, i.e. the applicability of the model not only for a certain domain, is fulfilled by most of the models.

Table 8: **Reasons for Selection of the Models per Graph Type.** The number of citations is taken from google scholar in the beginning of March 2023.

| Model | Reason(s) for Selection |
|---|---|
| *Models from Tab. 1* | |
| GenRecN (Sperduti & Starita, 1997) | cited often (783 times), historical importance |
| MagNet Zhang et al. (2021) | explicit addressing of graph property |
| GNN* Scarselli et al. (2009) | cited very often (5433 times) |
| GAT Velickovic et al. (2018) | cited very often (5705 times), commonly used as basline |
| CapGNN Xinyi & Chen (2018) | cited often (172 times) |
| WL Morris et al. (2019) | cited often (914 times) |
| EGNN Gong & Cheng (2019) | cited often (195 times) |
| GRNN Ioannidis et al. (2019) | explicit addressing of graph property |
| AA-HGNN Ren et al. (2020) | explicit addressing of graph property |
| HAN Wang et al. (2019) | explicit addressing of graph property |
| *Models from Tab. 2* | |
| NDHGNN Tran & Tran (2020) | only model found for graph property |
| HyperGCN Yadati et al. (2019) | cited often (202 times) |
| HyperConvAtt Bai et al. (2021) | cited often (239 times) |
| LHCN Bandyopadhyay et al. (2020) | more straight forward than other models (simplicity) |
| HGNN Feng et al. (2019) | cited often (566 times), commonly used as baseline |
| AHGAE Hu et al. (2021) | only model found for graph property |
| HWNN Sun et al. (2021b) | explicit addressing of graph property |
| G-MPNN Yadati (2020) | explicit addressing of graph property |
| *Models from Tab. 3* | |
| EpiGNN La Gatta et al. (2020) | more straight forward than other models (simplicity) |
| DySAT Sankar et al. (2020) | cited often (254 times), commonly used as baseline |
| (WD/CD)-GCN Manessi et al. (2020) | cited often (229 times), can handle node attributes unlike DySat |
| GCRN Seo et al. (2018) | cited often (537 times) |
| DynGEM Goyal et al. (2018) | cited often (322 times) |
| EvolveGCN Pareja et al. (2020) | cited often (532 times) |
| RE-Net Jin et al. (2019) | only model found for graph property |
| DyHAN Yang et al. (2020) | only model found for graph property |
| DHAT Luo et al. (2021) | only model found for graph property |

| | |
|---|---|
| STHAN-SR Sawhney et al. (2021) | can handle edge attributes unlike HGC-RNN |
| HGC-RNN Yi & Park (2020) | more straight forward than other models (simplicity) |
| Hyper-GNN Hao et al. (2021) | cited often (33 times since 2021), more straight forward than other models (simplicity) |
| MGH Yan et al. (2020) | only model found for graph property that can handle all dynamics in DTR |
| *Models from Tab. 4* | |
| Know-Evolve (Trivedi et al., 2017) | cited often (337 times), commonly used as baseline |
| DyGNN Ma et al. (2020) | cited often (121 times), more recent and better performing than predessesor Know-Evolve |
| DyRep Trivedi et al. (2019) | cited often (301 times), commonly used as baseline |
| NeurTWs Jin et al. (2022) | recent model |
| PINT Souza et al. (2022) | recent model, better performace than TGN |
| TGN Rossi et al. (2020) | cited often (246 times), commonly used as baseline |
| HIT Liu et al. (2021) | only model found for graph property |
| *Models from Tab. 5* | |
| MGCN Lu et al. (2019) | explicit addressing of graph property |
| BipGNN Wang et al. (2020) | explicit addressing of graph property |
| BGNN He et al. (2019) | explicit addressing of graph property |
| GTN Yun et al. (2019) | explicit addressing of graph property, cited often(492 times) |
| DAGNN Thost & Chen (2021) | explicit addressing of graph property, cited often (51 times since 2021) |
| CTNN He et al. (2021) | explicit addressing of graph property |
| MPNN-R Yadati (2020) | explicit addressing of graph property |
| Hier-GNN Chen et al. (2022) | explicit addressing of graph property |
| *Models from Tab. 6* | |
| GCN Kipf & Welling (2017) | cited very often (21945 times) |
| Hyperbolic GNN Liu et al. (2019) | cited often (220 times), explicit addressing of graph property |
| KGIN Wang et al. (2021) | cited most (120 times) among models for this graph type, explicit addressing of graph property |
| HetG Zhang et al. (2019a) | cited often (775 times) |
| MXMNet Zhang et al. (2020a) | cited most (29 times) among among models for this graph type, explicit addressing of graph property |
| STGNN Kapoor et al. (2020) | often cited (148 times), explicit addressing of graph property |
| SBGNN Huang et al. (2021) | explicit addressing of graph property |
| JODIE Kumar et al. (2019) | often cited (342 times), explicit addressing of graph property |
| HVGNN Sun et al. (2021a) | explicit addressing of graph property |

