# OpenReview forum: "Graph Neural Networks Designed for Different Graph Types: A Survey"
_TMLR — Accepted by TMLR_

### Review · Reviewer_4uat · 2023-01-17

**Summary Of Contributions:**

With the increased popularity of GNN based approaches on data from different domains, the range of GNN models have grown significantly. This survey aims to examine different graph types and point out corresponding GNN models designed for such types. In addition, this work also points out graph types where there is a lack of GNN models. This survey provides a roadmap for novel and existing researchers working with specific types of graph data to identify relevant GNN models in the literature.

The authors also clearly listed the criterias for including specific work in this survey. These criterias are 1). Up-to-dateness of the model, 2). relevance of the model, 3). the generality of the model, 4). explicitness in addressing a given graph property and 5). simplicity of the model and the priority of selection is in the same order. The criterias help reader understand the significance of the selected model.

Lastly, I believe this survey can act as a guide for the GNN community to develop more models in areas which lacks GNN models and facilitate future research.

**Audience:**

Yes

**Broader Impact Concerns:**

There is no concern of ethical implications of the work to my knowledge as it is a survey paper examining existing work.

**Claims And Evidence:**

Yes

**Requested Changes:**

* suggestions and adjustments

1. following weakness 1, I believe adding the discussion on time complexity for each included model would strengthen the work but wouldn't be critical for my acceptance recommendation

2. following weakness 2, I think a discussion on the performance of the models selected would be useful but again it would mostly strengthen the work

3. following weakness 3, **I would request the authors to update the section on continuous-time dynamic graphs** with the references I included in mind. This is important towards my recommendation of acceptance as it will improve the Up-to-dateness of the survey on this section.

* minor suggestions and edits

Please update the write-up based on these minor suggestions, it wouldn't change my recommendation but it is very important for the clarity of the paper.

1. in introduction page 1 "or so-called "structures"  and on last line page 1  "solar power plants". Most quotation marks are formatted incorrectly in the paper, please check and update all quotation marks

2. "Remark. All combined static graphs can also be dynamic" on page 6. Does it mean that all types of static graph can also be dynamic? if so, this sentence was not clear on it.

3. "However, the continuous-time approach is much more compact in its representation but requires a local evaluation of the graph."  It is unclear what "a local evaluation" means.


**Strengths And Weaknesses:**

* Strengths

overall, I believe this survey can benefit the graph representation learning community through its systematic analysis of graph types and models. Overall, I would recommend acceptance with some edits required as listed in the weakness and request change section. I will first list the strengths below:

1. This survey systematically organizes different graph types and pointed out related GNN models to each type

2. The selection criteria for the inclusion of a given model is clearly discussed and the presentation of the survey overall is quite clear

3. the discussion around which areas lack GNN models are beneficial for the community and can be a guide for further future research

* Weaknesses

There are a few weaknesses and suggestions that the authors should improve upon:

1.  One important consideration of applying ML models on a given data type is scalability. Recently there is an increasing amount of work on scalable GNNs[1]. To improve the usefulness of the survey, clearly listing the time complexity for each model in the tables would help readers understand the scalability aspect. This can also facilitate future research especially if existing models are expensive in complexity.

2. The performance of a given model on a graph type should also be a selection criteria for model inclusion as it will point readers to the state-of-the-art model (as of the writing of the paper).

3. There is a large amount of recent work designing GNNs for continuous time dynamic graphs. The statement such as "the small number of models for continuous-time graphs" on page 22 should be reconsidered. I also hope the authors would include some of these recent papers and update the section on continuous-time dynamic graphs.  Some recent work include [2],[3],[4],[5]. In particular, [3] is one of the first work to design methods for signed continuous time dynamic graphs and should be included in the survey.


References:

[1]. Ding et al., Sketch-GNN: Scalable Graph Neural Networks with Sublinear Training Complexity, NeurIPS 2022

[2]. Luo et al., Neighborhood-aware Scalable Temporal Network Representation Learning, LOG 2022 Conference

[3]. Raghavendra et al., Signed Link Representation in Continuous-Time Dynamic Signed Networks

[4]. Jin et al., Neural Temporal Walks: Motif-Aware Representation Learning on Continuous-Time Dynamic Graphs, NeurIPS 2022

[5]. Souza et al., Provably expressive temporal graph networks, NeurIPS 2022

---

> ### Author Response · Authors · 2023-02-15
> **Answer to first review**
>
> Dear Reviewer 4uat,
>
> Thank you for your very helpful review and the short review time!
>
> We have included your minor editing suggestions and made changes in the paragraph on continuous-time dynamic graphs according to your suggestions in weakness 3. Please see our paper's new version and detailed comments below.
>
> To make it more comfortable for you to find our changes in the text, we highlighted them with color. The new version also includes smaller changes, as suggested by the other reviewers. Please let us know if you agree to the changes.
>
> Best regards,
> the authors
>
> Regarding weaknesses 1 and 2:
>
> Discussing the time complexity and performance of the models are important and necessary investigations regarding the choice of an appropriate model and thus are surely beneficial for the reader. For this reason, we have already discussed the addition of these topics among the authors. However, since different graph types yield different complexities in the models handling them, the time complexities we could give would not be comparable. Similarly, the performance of the models is challenging to compare since different graph types yield different data that the models are evaluated on. Nevertheless, since you and reviewer BMQA both listed these points as a weakness of our paper, we would like to suggest adding another table to the appendix (some of the tables in the main part already cover the complete page and would become unreadable with more information, we believe). It would contain the time complexity and performance as pointed out in the papers for each model. And since reviewer BMQA would like us to give links to implementations of the code, we would also add these here (if public repositories exist).
>
> Regarding weakness 3:
>
> We agree that the amount of continuous-time dynamic GNNs is rapidly growing, and our usage of 'few models' was misleading. We want to state that models exist for a few graph types in the continuous-time dynamic case compared to the discrete case. We have adapted the text accordingly.
>
> The models you suggest adding are very interesting.
>
> We decided to add the model from the paper [5] because of its higher expressivity than TGN. We also changed the ability of TGN such that it can handle dynamic edge attributes. Additionally, we added a footnote since the handling of edge attributes in TGN is only stated to be a possible extension, while in [5], it has been used.
>
> We included [4] in our paper because of its particular architecture that differs from the other models. Additionally, it covers are new graph type. Thanks a lot for the valuable suggestion.
>
> The model in the paper [3] performs well and is described well, which makes it very useful for a survey. However, the model explicitly handles sign graphs, a particular case of heterogeneous graphs, where the number of link types is 2. Also, the dynamic that the model can handle is restricted to strictly edge-growing graphs. For the purpose of this survey, the model, therefore, does not add anything to the models already in the table, and we decided not to include it.
>
> Similarly, the model in [2] can only handle growing edge dynamic graphs and perform edge prediction. This does not add information to the table, so we decided not to include it.

---

> > ### Comment · Reviewer_4uat · 2023-02-20
> > **Reply to Author Comment**
> >
> > Dear authors,
> >
> > Thank you for addressing my concerns on the paper. Also the edits on the continuous-time dynamic graphs section has reflected recent development in this field sufficiently. I will take these edits into the consideration.

---

### Review · Reviewer_wySu · 2023-01-24

**Summary Of Contributions:**

This paper is a survey that categorized existing GNNs according to what graph types they could handle. First, they gave mathematical definitions of various graph types and basic GNN architectures. Then, for each graph type, they gave representative GNNs that could handle it. Based on these classifications, the paper identified graph types that many GNNs supported and ones that few GNNs supported and discussed the reasons for the differences.

According to the authors, the contributions of this paper are as follows (P2, Section 1):
1. gave a comprehensive definition of the various graph types
2. classified existing GNNs by graph types they can handle
3. related GNNs to each other based on their architectures
4. identified graph types that current GNN research is missing

**Audience:**

Yes

**Broader Impact Concerns:**

I have no major concerns about broader impacts.

**Claims And Evidence:**

Yes

**Requested Changes:**

[R-1] P1, Section 1: ,,structure'' -> I do not think using such quotations in English papers is common. "structure" or $\textit{structure}$ are more suitable.

[R-2] P3, Section 3.1:

>In this sense, a directed hypergraph is a directed graph that simultaneously is a hypergraph

In Definition 3.1, hypergraphs are described as elementary. However, this sentence described them as a composition of elementary objects. Therefore, they need to look more consistent.

[R-3] P4, Figure 1: I would like to confirm the source of the dog image.

[R-4] P5, Definition 3.3: In the definition of growing, $\mathcal{E}\_i \subseteq \mathcal{E}\_{i+1}$ implicitly assumed $\mathcal{V}\_i \subseteq \mathcal{V}\_{i+1}$. Therefore, it is weird that these conditions were connected by the "or" expression. The same is true for the definition of shrinking.

[R-5] P6, Definition 3.6: I would like to clarify how the paper used the following terms: structure, topology, and semantics. It looks strange to call graph properties such as complete, regular, bipartite, and so on "semantic ." Also, I could not understand the following sentence in P18:
> Although semantic graph properties typically do not explicitly affect the graph's topology, [...]

I want to know what the authors intended to mean by a graph's topology -- I think that the properties such as complete are something that I imagine the topology of the graph:

[R-6] P6, Definition 3.6: 7. recursive and 10. hyperbolic were not well defined. Therefore, it is difficult to understand these concepts from these descriptions alone. I suggest the authors guide the readers to the reference therein.

[R-7] P6, Definition 3.6, 8: Edges in $H$ correspond to ... -> $G$

[R-8] P7, Definition 3.7, 4: The notations $v\in e$ in 1 and $e\subseteq \mathcal{V}$ implied that the graph is interpreted as a hypergraph. It is better to state this explicitly.

[R-9] P7, Definition 3.7, 7: random -> randomly

[R-10] P7, Definition 3.7, 8: It would be difficult for someone unfamiliar with the concept of metapaths to understand its meaning from this definition alone. It is preferable to provide appropriate definitions or references.

[R-11] P7, Definition 3.7, 1: indegree -> in-degree

[R-12] P8, Eq. (2): Strictly speaking, GAT cannot be formulated in the form Eq. (2). $\mathrm{att}$ corresponds to $a_{ij}$ of an equation in P.9. However this function depends not only on $x_u$ and $x_v$ but also on all $x_i$'s for $i\in \mathcal{N}(v)$. Nevertheless, it would be too complicated to describe this in a rigorous mathematical manner. Therefore, I think it is sufficient to explain it as a supplement to Eq. (2).

[R-13] P9, Section 3.2.4: I think it is better to use a more special character to denote the set of real numbers (e.g., $\mathbb{R}$)

[R-14] P9, Section 3.2.4: $\boldsymbol{X}$ -> $\boldsymbol{X}^{(0)}$

[R-15] P10, Section 4.1:
> The number of GNNs for simple graphs [...], so the following table does not list all of them [...]

I think it is better to write explicitly which table this sentence refers to (Table 1?). The same applies to Table 2 in Section 4.2.

[R-16] P10, Section 4.1: The authors referred to GenRecN as an example of GNN for directed edges. Certainly, this paper has 723 citations (google scholar, 2023/1/14) and is a very important study. However, this paper was published in 1997 and is not positioned in the GNN studies starting from the late 2000s. Since one of the criteria for selecting GNN models in this paper is that they are relatively new, I would recommend mentioning other models, such as DimeNet [1]?

[R-17] P10, Section 4.1: Add reference to HAN.

[R-18] P13, Section 5: [...], Def. 3.4 The first [...] -> [...] Def. 3.4 The first [...] -> [...]

[R-19] P13, Section 5 and P16, Section 5.2: This paper explained that continuous-time graphs/GNNs were more compact than discrete-time graphs/GNNs. However, I think this is not a matter of discreteness or continuity of dynamic graphs. Rather, it is a matter of representation of graphs. There are two ways to represent a dynamic graph: taking a snapshot of graphs at each time step or saving the changes of graphs. The latter is expected to be more compact representations. While discrete-time graphs can take both approaches, continuous-time graphs can only take the latter approach. Therefore, a compact representation is not an advantage of the continuous-time graph but rather an advantage of the representation method.

[R-20] P18, Section 6: Over-smoothing is a technical term with special meaning in the context of GNNs, so a reference should be provided, e.g., [2].

[R-21] P21, Section 8: Moreover, [...] how they handle these. -> them.

[1] https://openreview.net/forum?id=B1eWbxStPH

[2] https://ojs.aaai.org/index.php/AAAI/article/view/11604

**Strengths And Weaknesses:**

**Strengths**
1. Few survey papers focused on the graph topology (semantic) and classified GNNs according to which topology they can handle (Section 6)
2. The paper not only listed GNNs that support each graph type but also discussed how much and why GNN research for the type was active or not.
3. The paper is well-organized and clearly written.

**Weaknesses**
1. There is room for improvement in the description of graph-type definitions.
2. Contributions 1 and 3 above could be stronger in terms of novelty.


**Soundness**

[S-1] I think the above contributions are well-supported.

[S-2] Contribution 1 is discussed in Section 3. Specifically, graph types such as elementary properties (simple graph/hypergraph), temporal properties (static/discrete-time dynamic/continuous-time dynamic graph), and semantic properties (e.g., bipartite/complete.) This paper considered graph types for each property, such as undirected/directed, node-/edge- attributed, and node-/edge- heterogeneous. In addition, special examples of their combinations were discussed, such as knowledge graphs, multi-relational graphs, and multiplex graphs. These definitions are reasonable, although there are some minor suggestions (see Requested Changes).


[S-3] Contribution 2 is discussed in Sections 4--7: Section 4 for simple graphs and hypergraphs, Section 5 for static and dynamic graphs, Section 6 for graphs with semantic properties, and Section 7 for combined graphs. I am afraid I have yet to be able to check all GNN models this paper referred to. However, as far as I know, there were no major errors in the paper's classification of GNNs.

[S-4] Contribution 3 is discussed in Section 3.2, although it is not explicitly stated in the text. Since the contents mostly relied on Bronstein et al. (2021), this part is not the contribution of this paper.

[S-5] Contribution 4 is discussed in Section 8. I agree that for node-heterogeneous static graphs, graphs with duplicated nodes/edges, and continuous-time dynamic graphs, few GNNs support these graph types. Also, the reasons for this phenomenon were reasonable.

**Novelty and significance**

[N-1] For Contribution 1, giving mathematical definitions of the graph types was preferred because the definitions of these graph types often vary from paper to paper. It certainly increased the readability of the paper. On the other hand, the graph types were not new in themselves. Also, this unification of terminology was mainly for the preparation of the following sections (especially, Contribution 2). Therefore, novelty is limited on its own. Nevertheless, this classification of the graph types serves as a foundation on which subsequent studies will be developed.

[N-2] For Contribution 2, I agree with the authors' claim that although there were many GNN surveys for individual graph types (or some of them), few studies comprehensively classified them based on graph types. In addition, surveys of GNNs focusing on semantics are rare. From these points, I think we can recognize the novelty in this respect.

[N-3] For Contribution 3, since the discussion follows that of Bronstein et al. (2021), I do not think this part is novel.

[N-4] For Contribution 4, the discussion is novel. To the best of my knowledge, little literature has analyzed the relationship between graph types and the amount of GNNs models that supported them. However, I have several questions regarding the discussion on continuous dynamic graphs (see Requested Changes).

---

> ### Author Response · Authors · 2023-02-15
> **Answer to first review**
>
> Dear Reviewer wySu,
>
> Thank you for your very helpful review and the short review time!
>
> We have included your requested changes. Please see the new version and our detailed comments below (where needed).
> To make it more comfortable for you to find our changes in the text, we highlighted them with color. The new version also includes smaller changes, as suggested by the other reviewers. Please let us know if you agree to these changes as well.
>
> Kind regards,
> the authors
>
> Soundness 4:
> You mention that contributions 1 and 3 could be stronger in their novelty: The first contribution is, as you said, the comprehensive summary of the definitions of the various graph types. Since we are not reinventing these definitions but only putting them together to bring order to the concepts and to give an overview for contribution 2, we see no need for novelty. Regarding contribution 3, you say "that it is discussed in Section 3.2, although this is not explicitly stated in the text. Since the contents of Section 3.2 mostly rely on Bronstein et al. (2021), this part is not a contribution of this paper but instead a service to the reader. However, we did not make that clear enough, so we formulated it more clearly in the introduction. The actual contribution does not refer to Section 3.2. It refers to the discussions in the main chapters (4-7) in which we discuss the GNNs that appear in the tables regarding their functionality and GNN architecture. Please let us know if this clarifies our point for you.
>
> R2: We understand the confusion. In the elementary graphs, we have defined the simple graph and the hypergraph as two different elementary building blocks. The example of the directed hypergraph was only meant to illustrate that these building blocks can be combined. To emphasize this, we added the word "simple" to the directed graph. In the following sentence of the paper, it is directly stated that these are, of course, not the only combinations. However, one can extend these different elementary graphs with the help of further graph properties, which are listed afterward. Examples of these combined graph types graphs (elementary graph + graph properties) that are common in the literature and important for GNNs are given in Def. 3.5.
>
> R3: We previously had the logo of our working group in the figure and have replaced it with a self-shot photo of the dog of one of the authors to fulfill the requirements of the blind review. In the final version, we will change it back to our logo if you agree. Otherwise, we can state the origin of the foto.
>
> R4: Unfortunately, the implicit assumption that if the edge set grows, the node-set must grow is false because the graph can have new connections between already existing nodes. The "or" we used in the paper is a mathematical "or" and not an exclusive "or". I.e., it can grow both, but it does not have to.
>
> R5: The graph structure is defined only by the mathematical objects that make up the graph, i.e., node, edge, attribute, or node/edge type sets. The so-called structural properties can be deduced from these sets alone, e.g., whether the graph is directed, attributed, heterogeneous, or a multigraph. For example, the node set of a multigraph is mathematically different from a simple graph. On the other hand, the semantic properties do not affect mathematical representation, e.g., the graph's adjacency matrix. These instead result from an interpretation of the graph, e.g., whether the graph is cyclic, a tree, etc. That is, the semantic properties of a graph do not affect the graph structure in that it would change anything about the node edges or attribute set. We initially used the word 'topology' as equivalent to 'structure'. We realize this needs to be corrected and clarified. We actually do not need the word 'topology', so we do not use it anymore.
>
> R6: We realized the definition of a hyperbolic graph is unnecessary, and since our prior definition required the reader to read the complete reference, we deleted it. The definition of a recursive graph was indeed confusing because it should have been the definition of a recursive hypergraph. We changed it accordingly and pointed to the original definition and more intuitive visualization.
>
> R8: The graph defined in Definition 3.7.4 is not interpreted as a hypergraph. It is a subgraph of the original graph defined by a subset of the node and edges of the original graph. There was a mistake in defining the induced subset of edges, and we updated the definition. Thank you for noticing it!
>
> R10: We realized we do not actually use the definition of a metapath in the survey and therefore decided to leave it out.
>
> R20: We totally agree. Maybe, this is a misunderstanding, but we believe that we have always only talked about a graph's time dynamic representation. We also specify the representations in Def. 3.4. Could you please clarify where we have written that the graph itself is more (or less compact) and not its representation?

---

> > ### Comment · Reviewer_wySu · 2023-02-21
> > **Response to authors' comments**
> >
> > I thank the authors for answering my review comments sincerely. I would like to respond authors' comments one by one.
> >
> > [S-4] OK
> >
> > [R-2] OK
> >
> > [R-3] It is OK for me to use the logo after the authors can reveal their identity if using the logo is permitted by the working group.
> >
> > [R-4] In my understanding, $\mathcal{V}\_{i} \subseteq \mathcal{V}\_{i+1}$ includes the case $\mathcal{V}\_{i} = \mathcal{V}\_{i+1}$. That is, the node set does not have to change. With that being said, after reading the authors' response, I come to think that keeping the original definition is OK because if we write $\mathcal{E}\_{i} \subseteq \mathcal{E}\_{i+1}$ and do not write $\mathcal{V}\_{i} \subseteq \mathcal{V}\_{i+1}$, some readers may mistakenly think that $\mathcal{V}\_{i} = \mathcal{V}\_{i+1}$ is assumed.
> >
> > [R-5] I understand how the authors distinguish "structure" and "semantic". Because I do not think these definitions are not obvious to all readers, I would recommend explaining how these terms are used in this paper.
> >
> > [R-6] OK
> >
> > [R-8] OK
> >
> > [R-10] OK
> >
> > [R-16] Unless I am missing some information, there was no discussion about adding DimeNet as a reference. Although I am not very keen on adding it, I would like to clarify whether it was done on purpose. Also, I appreciate that the authors have added MagNet as an example of GNN for directed graphs.
> >
> > [R-17] Wang et al. (2019) -> (Wang et al., 2019) (use citep)
> >
> > [R-20] I am sorry for the confusion. I also intended the representation of graphs. For example, we can represent a discrete-time graph using both the update-based and snapshot-based representations. If the graph has a small number of updates, we expect that the latter can represent the graph more compactly than the former. In other words, whether a graph can be represented compactly is a matter of what kind of representation we use, not whether the graph is continuous or discrete.
> >
> > **Additional Comments**
> > - Section 4: table 1 -> Table 1 (capitalize)
> > - Section 4.1: table 2 -> Table 2 (capitalize)

---

> > > ### Author Response · Authors · 2023-02-21
> > > **Answer to**
> > >
> > > Dear wySu,
> > >
> > > Thank you for the quick answer and the beneficial feedback. We made further changes in the document and uploaded a new version. Below, you can find the answers to your comments.
> > >
> > > Kind regards,
> > >
> > > the authors
> > >
> > > R5: We added an explanation for structural and semnatic graph properties on page 3 (marked in blue). We believe this makes the difference between these properties clearer for the readers.
> > >
> > > R16: We are sorry, we forgot to mention that in our last answer. For the survey, DimeNet does not appear suitable to us. Although it is significantly newer, simpler and models directed graphs, it is explicitly designed for molecular applications. Therefore, we decided to include MagNet instead because of its more general architecture.
> > >
> > > R17: corrected
> > >
> > > R20: We are very sorry, but we do not understand what you mean by discrete and continuous graphs. In our understanding, these are only two different representations that can represent the same dynamic graph. By discrete graph, do you mean that the available data is in discrete representation?
> > >
> > > Additional comments: corrected

---

> > > > ### Comment · Reviewer_wySu · 2023-03-02
> > > > **Response to authors' comments**
> > > >
> > > > Dear authors
> > > >
> > > > Thank you for your response and I am sorry for my late response. I missed your additional questions about R-20.
> > > >
> > > > [R-5] OK
> > > >
> > > > [R-16] OK
> > > >
> > > > [R-17] OK
> > > >
> > > > [R-20] By the discrete-time graph, I meant a sequence of graphs $G_i$ indexed by a discrete variable $i$ (e.g., integers $i = 1, 2, \ldots$). On the contrary, by the continuous-time graph, I meant a sequence of graph $G_t$ indexed by a continuous variable $t$ (e.g., non-negative real values $t\geq 0$)
> > > >
> > > > Additional Comments
> > > >
> > > > P3: Semantic Properties -> Semantic properties
> > > >
> > > > Best,
> > > > Reviewer wySu

---

> > > > > ### Author Response · Authors · 2023-03-02
> > > > > **Response to R-20 of wySu**
> > > > >
> > > > > Dear reviewer wySu,
> > > > >
> > > > > Thank you for clarifying the last open question [R-20]. Indeed, we forgot to emphasize that the time steps in defining the discrete-time representation are not necessarily equidistant. We now emphasize it in the definition correspondingly. The difference to the continuous-time representation is that in the discrete-time representation, the graph is characterized as a sequence of graph snapshots at (arbitrary) time steps. In contrast, in the continuous-time representation, the graph contains a (possibly empty) start graph and a set of occurring events over time.
> > > > >
> > > > > Best Regards,
> > > > > the authors

---

> > > > > > ### Comment · Reviewer_wySu · 2023-03-02
> > > > > > **Response**
> > > > > >
> > > > > > Dear Authors
> > > > > >
> > > > > > Thank you for further clarification about [R-20]. I think I understand the authors' intention.
> > > > > >
> > > > > > Best,
> > > > > > Reviewer wySu

---

### Review · Reviewer_BMQA · 2023-02-02

**Summary Of Contributions:**

The paper provides a survey of Graph Neural Networks (GNNs) with a focus on different graph types and properties and the GNNs designed for the corresponding categories. In particular, Section 2 and Section 3 introduce the basic knowledge of graphs and GNNs. The following sections (4, 5, 6, and 7) summarize existing GNN research on structural graphs, dynamic graphs, semantic graphs, and combined graphs.

**Audience:**

Yes

**Claims And Evidence:**

No

**Requested Changes:**

Please refer to the comments in weakness.

**Strengths And Weaknesses:**

Strengths:

1. The motivation of the paper is great since it tries to summarize the current research on GNNs from the data perspective. Essentially, various graphs of types and attributes might need a customized GNN design. The survey provides a more comprehensive view compared with existing survey papers.

2. The paper tries to provide a high-level categorization of all kinds of graphs and summarizes representative GNN models for each category. Some model designs are discussed and analyzed.

Weaknesses:

1. The writing of the paper needs significant improvement in terms of language, grammar, and logic.

2. The taxonomy of graphs in Section 2 is a bit unclear and confusing. Although it is mentioned that all basic definitions concerning graph types and their structural properties are taken from Thomas et al. (2021), there is a lack of discussion on why this taxonomy is reasonable and suitable for the discussion of GNNs. For instance, what are the intrinsic differences between "Structural graphs" and "Semantic graphs"?

3. While the paper mentions that the cited models are selected according to their up-to-dateness, relevance, general applicability, explicit addressing of a specific graph type, and simplicity. It is unclear how these cafeterias are actually carried out. A significant portion of references is from arXiv or some less-known venues while only a small portion of papers is from well-known Machine Learning or Data Mining venues. For instance, to the best of my knowledge, there are many more representative models for directed graphs not being cited while the cited works are less known.

4. The paper summarizes many existing works, but there is a lack of deeper discussion on their advantages, disadvantages, connections, potential, and empirical performance.

5. It would be better to provide a summary of implementations of cited works, i.e., adding links to the table.

Other minor comments:
1. From what is described in section 3.2.3, the differences between spatial and spectral convolution are unclear.

2. It is suggested to adjust the wording to avoid lines with one a few words. This can make the space compact and save more space for deeper discussions.

---

> ### Author Response · Authors · 2023-02-15
> **Answer to first review**
>
> Dear Reviewer BMQA,
>
> Thank you very much for your timely review and helpful feedback!
>
> We have already included changes to some of your minor points in a new version. The new version also includes smaller changes, as suggested by the other reviewers. Please let us know if you agree to these changes as well.
>
> To make it more comfortable for you to find our changes in the text, we highlighted them with color.
> In the following, we would like to address all your points regarding the weaknesses.
>
> Best regards,
> the authors
>
> Weakness 1: We are sorry that our writing style was unclear or incorrect for you. We have already made quite an effort to use appropriate vocabulary and grammatical rules and checked them several times with the help of spell-check and grammar tools. The other reviewers have also pointed out unclarities, which we have corrected. Additionally, we have proofread the paper once again and made several modifications. Since the other reviewers were okay with our writing style, we are still determining which other changes you require for a pleasant reading experience. Please tell us more explicitly which linguistic errors are unacceptable or where the manuscript needs clarity. We will change the text accordingly.
>
> Weakness 2: Only the structural graph properties were taken from the paper by Thomas et al. Since these already cover many but not all graph types important for our survey, we have extended the graph types in the survey by the other missing semantic properties. We have tried to emphasize this better in the paper and added a short explanation for their importance in the context of GNNs.
>
> Weakness 3: It is correct that, for reasons of space and readability, we have not explicitly mentioned in the tables which criterion made us select each model. It is also true that many models cited are from arxive or less known venues. This reflects the fast pace at which GNN models are published and that the paper's main goal is to show which graph types can be handled by GNN models. However, we agree that the model GenRecN was old and have therefore added MagNet, a spectral model for directed graphs presented at Neurips in 2021. We have also replaced two other models due to the comments of reviewer 4uat. Thank you for these helpful hints! We believe our literature research is sound for the other graph types. However, we would be happy if you have other explicit suggestions for which models can be replaced with others published in peer-reviewed or higher-ranked journals.
>
> Weakness 4: We have already discussed the addition of time complexity and performance of the models among the authors. However, since different graph types yield different complexities in the models handling them, the time complexities we could give would not be comparable. Similarly, the performance of the models is challenging to compare since different graph types yield different data that the models are evaluated on. We are also not convinced of the benefit of discussing the advantages/disadvantages of models made for different graph types since this would lead to a discussion of the most suitable graph type rather than a suitable model for a graph type. Nevertheless, since you and reviewer 4uat both listed similar points as a weakness of our paper, we would like to suggest adding another table to the appendix (some of the tables in the main part already cover the complete page and would become unreadable with more information, we believe). It would contain the time complexity and performance as pointed out in the papers for each model. Would that address your concerns?
>
> Weakness 5: Do you mean links to GitHub repositories for the code of each model? If so, we do not see a significant advantage for the readers because we don't focus on explaining each model in detail. If the reader wants to apply the model, a closer look at the paper is required either way. In addition, not for all papers mentioned in our survey code is provided. However, if you disagree, we can add the links to the table in the appendix as proposed to address weakness 4. Please let us know what you think.
>
> Minor comments:
>
> It is indeed difficult to explain a spectral convolution as short as we would like for this survey. We have added two sentences to the description. Please let us know if this is sufficient or if we should add a longer paragraph.
>
> We changed the wording of paragraphs that ended with just one or two words (see blue changes that end with minor 2 BMQA). We hope this makes the paper more compact. Unfortunately, due to the other changes and us coloring the changes in blue and striking out some lines, at the moment the manuscript looks less compact and ordered. If all reviewers agree to the changes and we can remove the color etc., we will attend to this point again.

---

> > ### Comment · Reviewer_BMQA · 2023-03-01
> > **Further comments**
> >
> > Dear authors,
> >
> > Thanks for the response and revision. I still have major concerns and comments about point 3 and point 4 mentioned in my original review:
> >
> > > Point 3: "While the paper mentions that the cited models are selected according to their up-to-dateness, relevance, general applicability, explicit addressing of a specific graph type, and simplicity. It is unclear how these cafeterias are actually carried out. A significant portion of references is from arXiv or some less-known venues while only a small portion of papers is from well-known Machine Learning or Data Mining venues. For instance, to the best of my knowledge, there are many more representative models for directed graphs not being cited while the cited works are less known."
> >
> > Further comment: I am still not convinced that the revision provides a comprehensive survey of existing research. To be honest, I think the current selection of papers is a bit random. While MagNet is included now, there are many related references in the paper of MagNet that are not mentioned in this submission. The same concern applies to other graph types. Moreover, it is unclear whether the selected papers reflect state-of-art research. I would like to suggest the authors do a comprehensive literature survey and improve the revision.
> >
> > > Point 4: "The paper summarizes many existing works, but there is a lack of deeper discussion on their advantages, disadvantages, connections, potential, and empirical performance."
> >
> > Further comment: I am not asking for the comparisons among all models which is obviously impossible since they are not comparable because the graph types are different. However, as a survey paper, beyond listing the name of the papers, it will be necessary to discuss the difference, connections, advantages, disadvantages, potentials, and even performance if possible within each category of GNNs for specific types of graphs. This can help the readers get a better understanding of existing research, especially the state-of-art.

---

> > > ### Author Response · Authors · 2023-03-02
> > > **Reply to further comments**
> > >
> > > Dear reviewer BMQA,
> > >
> > > Thank you for your further comments.
> > >
> > > To clarify why we selected which models for each graph type, we propose to discuss the selection criterion for each model in the text. We believe this will help to understand the number of models considered for a specific graph type.
> > >
> > > We do not aim to provide an extensive analysis of all available models for each graph type. Instead, we focus on identifying which graph types are currently covered by existing models. Therefore, we think that discussing multiple models for each graph type is beyond the scope of this paper. Note that such an investigation would involve more than 50 distinct discussion sections according to the distinguished graph types for which GNN models exist.
> > >
> > > To address your concerns, we propose to add a discussion of multiple GNNs for the most common graph types in addition to the explanation of respective selection criteria. Would that address your concern appropriately?
> > >
> > > During our research for the survey, we identified the following graphs as the most common ones:
> > >
> > > static
> > > - simple graphs: node attributed (from a historical point of view, it is the graph type considered first and therefore often used. In addition, it can already represent a large number of use cases)
> > > - hypergraphs: undirected (the simplest form of a hypergraph, other types are typically more complex and therefore used less frequently)
> > >
> > > dynamic graphs:
> > > - DTR: heterogenous (since knowledge graphs take the form of heterogenous dynamic graphs, they are common in literature )
> > > - CTR: multi (they are a hot topic in research due to their high representation capacity of multigraphs and the efficient computation of ctr graphs)
> > >
> > > Best regards,
> > > the authors

---

> > > > ### Comment · Reviewer_BMQA · 2023-03-02
> > > > **Comments**
> > > >
> > > > Dear authors, thanks for your response. I will take these into consideration.

---

> > > > > ### Author Response · Authors · 2023-03-15
> > > > > **Added reasons for model choices to manuscript**
> > > > >
> > > > > Dear reviewer BMQA,
> > > > >
> > > > > we have added a table with the reasons for each of our model choices to the appendix of the manuscript. We discarded the idea of including the reasons in the text since we noticed that the readability strongly declined.
> > > > > We therefore just added a remark corresponding to the table in the text.
> > > > >
> > > > > We also removed the blue color from the changes in the text that seemed to have consensus among the reviewers. The only blue color left is in referencing the new table.
> > > > >
> > > > > Kind regards,
> > > > > the authors

---

### Decision · Action_Editors · 2023-03-21

**Recommendation:** Accept with minor revision

**Comment:**

Two of the reviewers recommended to accept the paper and agreed that the manuscript was technically correct and relevant to the
TMLR community, especially to researchers working on graph machine learning. This survey offers an overview of different graph types and  models to tackle ML tasks involving such graphs.

The 3rd reviewer was leaning toward rejection mainly for two reasons: (i) they questioned the choice of references by the authors and asked to select more high-quality and representative references (ii) lack of discussion on their connections, advantages, disadvantages, potential, limitations, and future development.

The authors uploaded a last revision appending a table to the paper explaining the choice of each of the models they chose to refer to in the survey. They also argued that "*We do not aim to provide an extensive analysis of all available models for each graph type. Instead, we focus on identifying which graph types are currently covered by existing models. Therefore, we think that discussing multiple models for each graph type is beyond the scope of this paper.* "

I believe that the addition of the table addresses concern (i) and I think it is reasonable that fully addressing (ii) goes beyond the scope and objective of the paper.

I thus decided to accept the manuscript, I think this will be a useful reference for the graph learning community.

Here are a couple of minor points to address in the last revision:

- Unless I missed it, I think Table 7 is not referenced anywhere in the main paper.

- Def 3.2:

1) the notation f_i : x \to 0 seems ambiguous to me, maybe consider an alternative notation, e.g. f_i : x \to \{0\} would be technically correct, or f_i(x) = \{0\}.

2) "All multigraphs are written as the set Gm": I recommend using the alternative phrasing " The set of all multigraphs  is denoted as Gm" (as it was done in 4))

4) remove the "-" after "node"

- Def 3.3: 1) "notion" should be "notation"

- Def 3.7:
1) remove "(R11)"
2) where \tilde{A} **is** the adjacency



**Audience:**

Yes.

**Claims And Evidence:**

Yes.

---

> ### Author Response · Authors · 2023-03-29
> **Upload of camera-ready version**
>
> Dear action editors,
>
> we would like to thank you very much for the review process, we believe our paper benefited greatly!
>
> We uploaded a de-anonymized camera-ready version including your minor revision points. Regarding point 2, we noticed that we did not use the sets of all graphs of a specific type and deleted it everywhere in Def. 3.2.
>
> We also removed whitespace and moved tables around within subsections to get rid of whitespace that came from the review process.
> In the process of de-anonymizing we exchanged the picture of the dog in Fig. 1 with our group's logo and added acknowledgements.
>
> Lastly, we would like to ask, since we all contributed equally, is it possible to hint this not only in the pdf but also on this website?
>
> Kind regards,
>
> the authors